# The conserved transcriptional regulator CdnL is required for metabolic homeostasis and morphogenesis in *Caulobacter*

**Selamawit Abi Woldemeskel[1], Allison K. Daitch[1], Laura Alvarez[2]°, Gaël Panis[3]°, Rilee Zeinert[4], Diego Gonzalez[5], Erika Smith[1], Justine Collier[5], Peter Chien[4], Felipe Cava[2], Patrick H. Viollier[3], Erin D. Goley[1]***

**1** Department of Biological Chemistry, Johns Hopkins University School of Medicine, Baltimore, MD, United States of America, **2** Department of Molecular Biology, Umeå University, Umeå, Sweden, **3** Department of Microbiology and Molecular Medicine, Faculty of Medicine, University of Geneva, Geneva, Switzerland, **4** Department of Biochemistry and Molecular Biology, University of Massachusetts-Amherst, MA, United States of America, **5** Department of Fundamental Microbiology, Faculty of Biology and Medicine, University of Lausanne, Switzerland

° These authors contributed equally to this work.
* egoley1@jhmi.edu

**Data Availability Statement:** All RNA-seq and Tn-seq (https://www.ncbi.nlm.nih.gov/bioproject/?term=PRJNA586876), microarray (https://www.

## Abstract

Bacterial growth and division require regulated synthesis of the macromolecules used to expand and replicate components of the cell. Transcription of housekeeping genes required for metabolic homeostasis and cell proliferation is guided by the sigma factor $\sigma^{70}$. The conserved CarD-like transcriptional regulator, CdnL, associates with promoter regions where $\sigma^{70}$ localizes and stabilizes the open promoter complex. However, the contributions of CdnL to metabolic homeostasis and bacterial physiology are not well understood. Here, we show that *Caulobacter crescentus* cells lacking CdnL have severe morphological and growth defects. Specifically, *ΔcdnL* cells grow slowly in both rich and defined media, and are wider, more curved, and have shorter stalks than WT cells. These defects arise from transcriptional downregulation of most major classes of biosynthetic genes, leading to significant decreases in the levels of critical metabolites, including pyruvate, α-ketoglutarate, ATP, $NAD^+$, UDP-N-acetyl-glucosamine, lipid II, and purine and pyrimidine precursors. Notably, we find that *ΔcdnL* cells are glutamate auxotrophs, and *ΔcdnL* is synthetic lethal with other genetic perturbations that limit glutamate synthesis and lipid II production. Our findings implicate CdnL as a direct and indirect regulator of genes required for metabolic homeostasis that impacts morphogenesis through availability of lipid II and other metabolites.

## Author summary

To grow and divide, bacteria must accumulate precursor molecules to support duplication and expansion of cellular materials. One mechanism by which bacteria do this is by regulating the expression of genes whose products are important for production of these molecules. How gene expression is maintained or altered to support synthesis of appropriate

ncbi.nlm.nih.gov/geo/query/acc.cgi?acc=
GSE139873), and ChIP-seq (https://www.ncbi.nlm.
nih.gov/geo/query/acc.cgi?acc=GSE140134) data
are publicly available in NCBI.

**Funding:** This work was funded by the National
Institutes of Health (NIH, https://www.nih.gov/)
through R01GM108640 (to EDG), R01GM111706
(to PC), R35GM130320 (to PC), T32GM08515
(training grant support of RZ), and T32GM007445
(training grant support of AKD) and by a Johns
Hopkins Discovery Fund (https://www.
hopkinsmedicine.org) grant to EDG. JC is funded
by Swiss National Science Foundation (SNSF,
http://www.snf.ch) grants 31003A_140758 and
31003A_173075 and PHV is funded by SNSF grant
31003A_182576. Research in the FC lab is
supported by MIMS (http://www.mims.umu.se),
the Knut and Alice Wallenberg Foundation (KAW,
https://kaw.wallenberg.org), the Swedish Research
Council (https://www.vr.se), and the Kempe
Foundation (http://www.kempe.com). The funders
had no role in the study design, data collection or
analysis, decision to publish, or preparation of the
manuscript.

**Competing interests:** The authors have declared
that no competing interests exist.

molecules to balance growth with nutrient availability is not fully understood. In this
paper, we describe the role of a regulator of gene expression called CdnL in maintaining
levels of molecules required for bacterial growth and reproduction. CdnL broadly impacts
the levels of genes required for most biosynthetic processes. CdnL's broad impact on tran-
scription has downstream consequences on growth rate, cell shape, and nutrient require-
ments for growth. We report that CdnL is particularly important for maintaining levels of
the amino acid glutamate and the cell wall precursor lipid II, each of which is critical for
supporting proper growth and cell morphology. Our results implicate CdnL as a broadly
conserved regulator of metabolic homeostasis, growth, and cell shape in bacteria.

## Introduction

In order to replicate, bacterial cells must synthesize enough macromolecules to double in size
using nutrients available in their environment. These macromolecules are used to build struc-
tural components of the cell such as the membrane and cell wall, to synthesize the multitude of
enzymes required for essential biochemical processes, and to duplicate genetic material. Tran-
scriptional control of housekeeping genes encoding factors that carry out these essential func-
tions contributes to maintaining metabolic homeostasis to support growth and development.
Transcription initiation of housekeeping genes is achieved by housekeeping sigma factors that
direct the core RNA polymerase (RNAP) to their promoter regions [1], but may be co-regu-
lated by other factors, including the CarD-like transcriptional regulator CdnL.

CdnL is broadly conserved in bacteria and is best-characterized in *Myxococcus xanthus* and
in *Mycobacteria* [2–6]. It localizes to promoter regions where the housekeeping sigma factor
resides, directly binds to the RNAP β subunit, stabilizes the open promoter complex, and is
required for transcription from rRNA promoters [2,3,5]. Recent work in *Mycobacterium
tuberculosis* using mutants of CarD (the CdnL homolog) unable to bind either RNAP or DNA
showed that CdnL can act as an activator or repressor of transcription depending on the stabil-
ity of the RNAP-promoter complex [7]. *M. tuberculosis* CarD is essential for growth in culture,
persistence in mice, and survival during genotoxic stress and starvation [2,4]. The CdnL
homolog in *Bacillus cereus* is upregulated during heat treatment and *B. cereus* spores lacking
CdnL do not recover as well as wild-type (WT) [8]. Additionally, the *Borrelia burgdorferi*
CdnL homolog LtpA is important for infection and is upregulated during cold shock [9,10].
These observations suggest that CdnL plays a role in stress response in diverse bacteria in addi-
tion to its role in mediating transcription of housekeeping genes.

In *Caulobacter crescentus* (hereafter *Caulobacter*), CdnL is not essential but is required for
normal growth [11]. Similar to *Mycobacteria* and *M. xanthus*, *Caulobacter* CdnL binds the β
subunit of RNAP and is required for transcription from an rRNA promoter [11]. In *Caulobac-
ter*, CdnL loss causes cell division defects, slow growth, and cold sensitivity [11]. Interestingly,
CdnL depletion in *M. xanthus* causes cell filamentation [6]. These last observations imply that
CdnL-dependent effects on gene expression are required to support proper morphogenesis in
*Caulobacter* and *M. xanthus*.

Bacterial morphology is maintained by the peptidoglycan (PG) cell wall, which must be
expanded and remodeled in a regulated manner to allow growth and division. It lies just out-
side the inner membrane and is a continuous, covalently-linked structure made of glycan
strands crosslinked together by peptide side chains [12]. In addition to giving the cell its ste-
reotyped shape, the cell wall provides physical integrity and prevents cell lysis due to turgor
pressure. Different shape features (e.g. cell length, width, or curvature) are specified by

spatially and temporally regulated synthesis and remodeling of the PG that is largely orchestrated by cytoskeletal proteins [12,13]. Though the details of PG synthesis and remodeling and its spatial regulation has been the subject of intense study for several decades, how PG metabolism is regulated in coordination with nutrient availability and metabolic status of the cell is incompletely understood.

Here, we investigate how CdnL affects metabolism, growth, and morphogenesis in the α-proteobacterium *Caulobacter crescentus*. *Caulobacter* undergoes stereotyped morphological transitions as it progresses through the cell cycle and during adaptation to a variety of stresses, making it an ideal organism to study bacterial morphogenesis [14]. Although it is known that *Caulobacter* CdnL is not required for viability, the consequences of its loss on *Caulobacter* transcription and physiology have not been previously characterized [11]. Here, we demonstrate that CdnL directly and indirectly regulates transcription of biosynthetic genes required for metabolic homeostasis that, in turn, supports proper growth, PG metabolism, and morphology in *Caulobacter*.

## Results

### Δ*cdnL* cells have growth and morphology defects

We became interested in *Caulobacter* CdnL through a screen for spontaneous suppressors of a dominant lethal mutant of the cell division protein FtsZ (called FtsZΔCTL) that causes lethal defects in PG metabolism [15]. Specifically, one of the FtsZΔCTL suppressors we identified carried a point mutation in *cdnL* (*CCNA_00690*) encoding an I42N missense mutation that affected CdnL protein abundance (S1 Fig). Though the suppression of FtsZΔCTL toxicity appears to be mediated indirectly, this observation prompted us to examine the effects of loss of CdnL on morphology and cell wall metabolism. To examine the role of CdnL in *Caulobacter* growth and morphogenesis, we deleted it in a clean genetic background and compared growth and morphology to WT cells. As previously reported [11], we found that Δ*cdnL* cells grew more slowly than WT (Fig 1A) and that they had pleiotropic morphological defects in complex PYE medium (Fig 1B and 1C). A prior description of the effect deleting *cdnL* on cell morphology noted only a cell division defect [11]. However, quantitative analysis of shape differences between WT and Δ*cdnL* cells grown in PYE using CellTool [16] revealed significant differences in two shape modes which approximately correspond to cell curvature (shape mode 2) and cell width (shape mode 3) (Fig 1C). Specifically, we observed that in PYE, *ΔcdnL* cells were wider and more curved than WT cells (Fig 1B, 1C and 1D). In these shape modes, Δ*cdnL* cells were also more variable in their range of values than WT, indicating less precise maintenance of shape in the mutant (Fig 1C). We observed occasional cell filamentation for Δ*cdnL* cells grown in PYE, but this was not the predominant shape defect. Finally, we observed phase-light "ghost" cells in Δ*cdnL* samples indicative of cell lysis and suggesting a lack of integrity in the cell envelope (Fig 1B, arrows).

To further characterize the Δ*cdnL* growth and morphology phenotypes, we grew WT and Δ*cdnL* cells in the defined media M2G or HIGG. Surprisingly, we found that M2G was unable to support growth of cells lacking CdnL, but that Δ*cdnL* cells grew in HIGG, albeit more slowly than WT (Fig 1A). A comparison of the components of M2G and HIGG media suggested that the missing nutrient in M2G that is required for growth of Δ*cdnL* cells might be glutamate. Consistent with this, addition of sodium glutamate to M2G (called M2GG with sodium glutamate added) supported growth of Δ*cdnL* cells to a similar rate as in HIGG. These data indicate that Δ*cdnL* cells are unable to synthesize sufficient glutamate to support growth and are now glutamate auxotrophs (Fig 1A). No other amino acids are included in either M2G or HIGG,

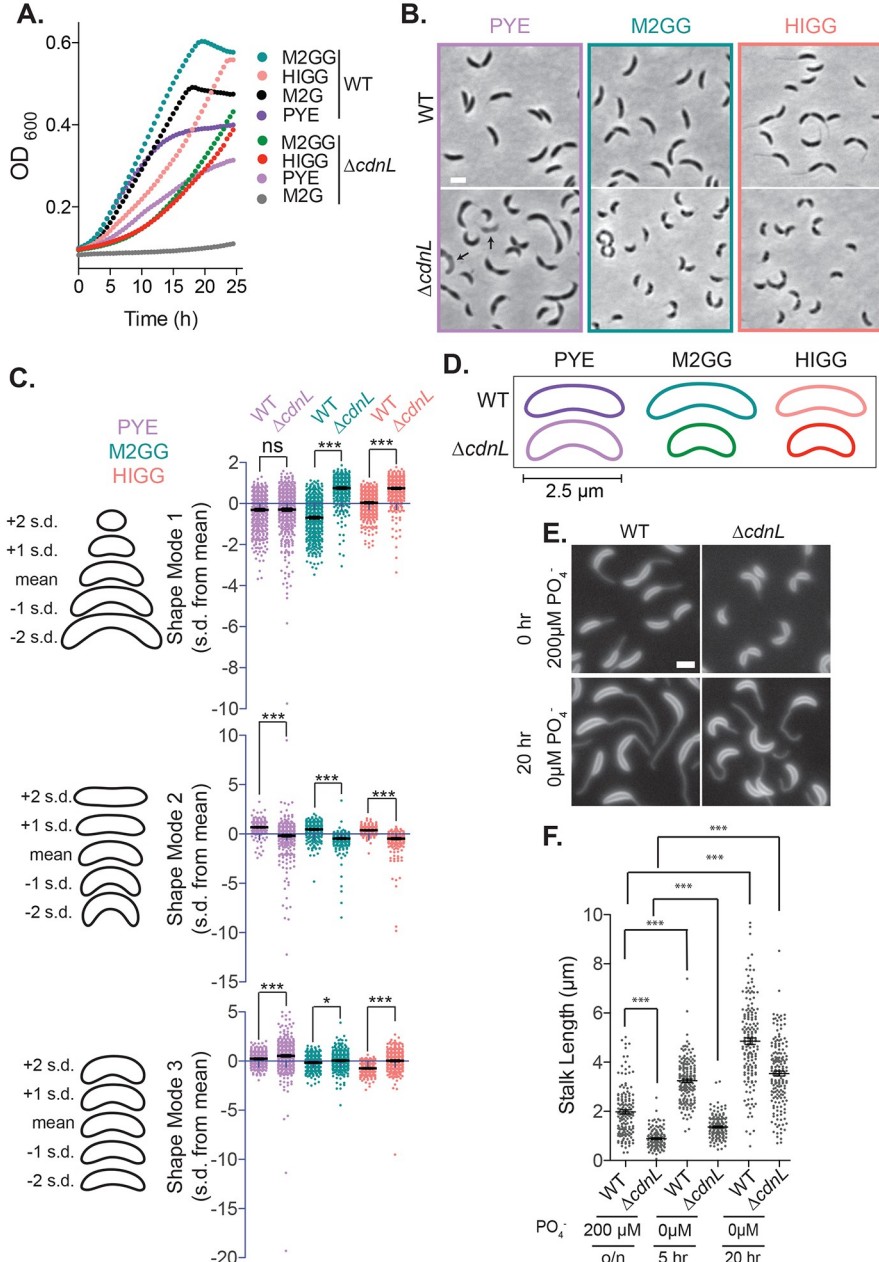

**Fig 1. Cells lacking CdnL have pleiotropic morphological and growth defects. A.** Growth of WT (EG865) and *ΔcdnL* (EG1447) cells in the indicated media. Experiments were performed in triplicate and mean is shown. **B.** Phase contrast images of WT and *ΔcdnL* cells grown in the indicated media. Arrows indicate lysed ghost cells. Bar = 2 μm. **C.** PCA of cell shape for WT and *ΔcdnL* shown in **B**. Scatter plots of normalized values (standard deviation (s.d.) from the mean value) for shape modes 1, 2 and 3 which approximately correspond to length, curvature and width. Contours on the left of scatter plots indicate shape of mean and 1 or 2 s.d. from the mean. **D.** Mean shapes of WT and *ΔcdnL* cells grown in the indicated media. **E.** Membrane staining of WT (EG865) and *ΔcdnL* (EG1447) cells with FM4-64 after growth in HIGG media with and without phosphate. **F.** Stalk lengths measured using ImageJ's [43] line tool. Mean and SEM included for plots in **C** and **F**. Statistical analysis in **C** and **F** performed using one-way ANOVA with Bonferroni's Multiple Comparison Test, n = 500 for each in **C**. n = 150, 127, 168, 155, 162, 172 from left to right in **E**. *** = P < 0.001, * = P < 0.05, ns = not significant.

suggesting a specific requirement for exogenous glutamate. CdnL protein levels are unchanged across media in WT cells (S1 Fig).

Similar to our observations with PYE, we found that Δ*cdnL* cells grow more slowly than WT in M2GG or HIGG and also had aberrant shape (Fig 1A and 1B). Δ*cdnL* cells were again hypercurved and wider than WT cells, but Δ*cdnL* cells were also consistently shorter than WT in each defined medium (Fig 1B, 1C and 1D). The polar stalk is a prominent morphological feature of *Caulobacter* cells and, like other features of cell shape, stalk biogenesis and elongation requires PG synthesis [17]. Since we observed stereotyped shape changes in cells lacking CdnL, we also compared stalk morphogenesis between WT and Δ*cdnL* cells. We found that in phosphate replete HIGG media (containing 200 μM PO$_4^-$), Δ*cdnL* cells had short stalks compared to WT cells (Fig 1E and 1F), suggesting that CdnL impacts all aspects of morphogenesis in *Caulobacter*. Stalks elongate in response to phosphate starvation in *Caulobacter* through a pathway that is distinct from developmentally regulated stalk morphogenesis [18,19]. Interestingly, Δ*cdnL* cells significantly elongated their stalks during phosphate starvation (Fig 1E and 1F) suggesting that Δ*cdnL* cells can still alter their morphology in response to phosphate starvation.

## The growth and morphological defects of Δ*cdnL* arise from loss of CdnL

Having established that deletion of *cdnL* causes growth and morphology defects, we sought to determine if these phenotypes were specifically attributable to CdnL loss. To test this, we constructed Δ*cdnL* strains with *cdnL* or, as a control, *cfp* expressed under control of the xylose-inducible P$_{xylX}$ promoter, and assessed their growth and morphology with and without xylose inducer (Fig 2). By immunoblotting with CdnL antisera, we found that CdnL was absent from Δ*cdnL* P$_{xylX}$-*cfp* cells in the presence or absence of xylose, as expected, and CdnL was cleared from Δ*cdnL* P$_{xylX}$-*cdnL* cells after 7 hours of growth in media without xylose (S1 Fig). We assessed how depleting CdnL affects *Caulobacter* growth by monitoring optical density of cultures of Δ*cdnL* P$_{xylX}$-*cdnL* and Δ*cdnL* P$_{xylX}$-*cfp* in PYE xylose or PYE glucose media for 24 h. We found that growth of Δ*cdnL* P$_{xylX}$-*cdnL* was indistinguishable from WT in the presence of xylose inducer and was indistinguishable from Δ*cdnL* or Δ*cdnL* P$_{xylX}$-*cfp* in the presence of glucose (Fig 2A). Δ*cdnL* P$_{xylX}$-*cfp* cells grew at rates similar to Δ*cdnL* in the presence or absence of xylose inducer. We also assessed growth of Δ*cdnL* P$_{xylX}$-*cdnL* cells in M2 minimal media and found a trend similar to that observed in PYE, with the exception that carbon source (i.e. glucose or xylose) influenced the growth rates of all strains tested independent of *cdnL* expression (Fig 2B and 2C). Interestingly, cells lacking CdnL showed moderate growth in M2 supplemented with xylose, suggesting that Δ*cdnL* cells can more readily synthesize glutamate using xylose, as compared to glucose, as a carbon source (Fig 2C, S2 Fig). Additionally, Δ*cdnL* P$_{xylX}$-*cdnL* cells grew better than Δ*cdnL* or the Δ*cdnL* P$_{xylX}$-*cfp* control in M2G or M2GG, suggesting that leaky expression of *cdnL* from the *xylX* promoter can support growth in defined medium to some extent (Fig 2B and 2C). Collectively, our growth analyses indicate that the slow growth and glutamate auxotrophy of Δ*cdnL* cells arise specifically from loss of CdnL and can be restored by expressing *cdnL* from an inducible promoter.

We next turned to the morphological defects observed for Δ*cdnL* cells and asked (1) if morphology is restored to WT upon expression of *cdnL* and (2) at what times post-depletion of CdnL specific aspects of morphology are altered. By imaging the CdnL depletion and control strains described above, we found that Δ*cdnL* P$_{xylX}$-*cdnL* cells grown in PYE with glucose for 24 hours or Δ*cdnL* P$_{xylX}$-*cfp* cells grown with xylose or glucose are morphologically indistinguishable from Δ*cdnL* (Fig 2, S2 Fig). Importantly, however, Δ*cdnL* cells expressing xylose-induced *cdnL* have WT morphology (Fig 2D, time 0), indicating that the morphological defects

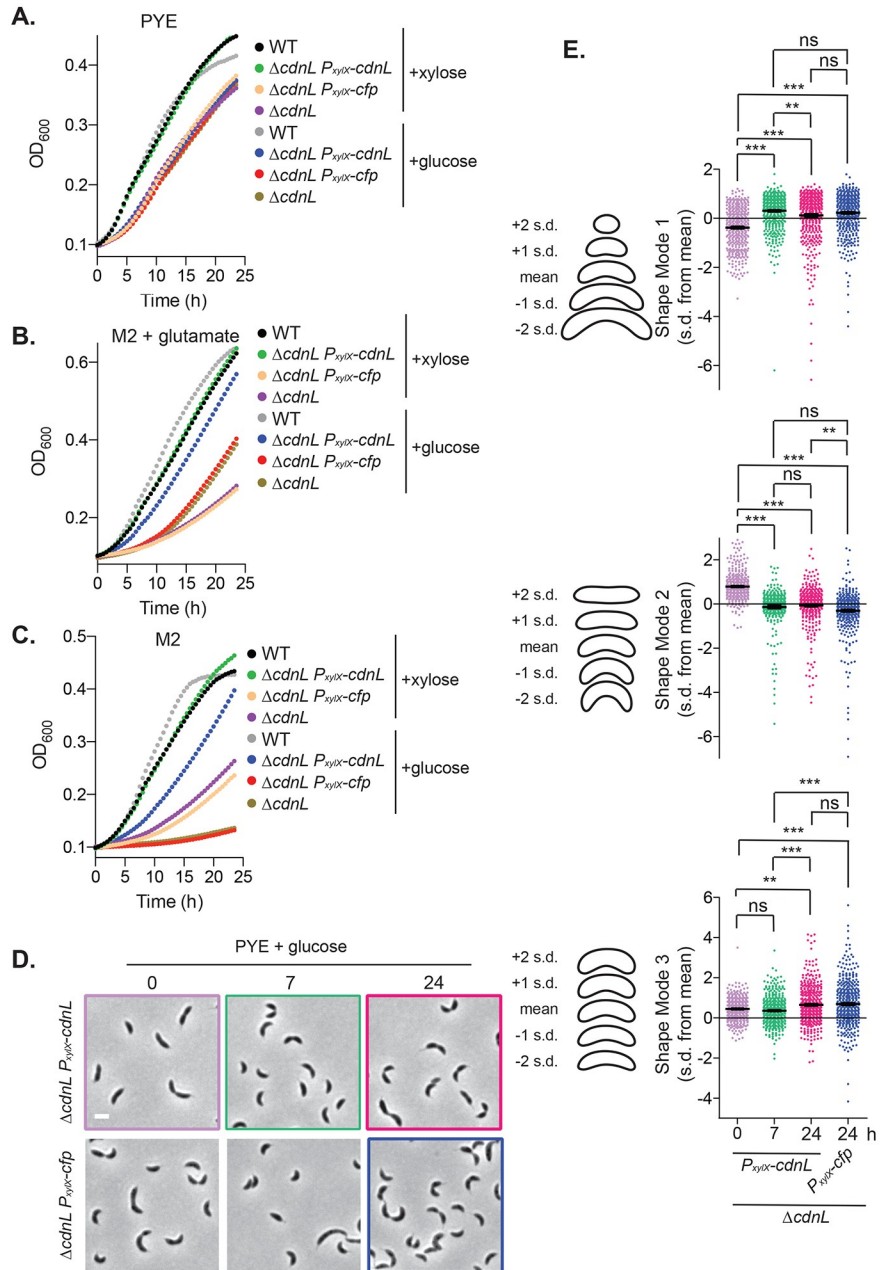

**Fig 2. Growth and shape defects of ΔcdnL are restored to WT by expression of cdnL. A-C.** Growth curves of WT (EG865), ΔcdnL (EG1447), ΔcdnL P$_{xylX}$-cdnL (EG1403), and ΔcdnL P$_{xylX}$-cfp (EG3136) cells grown in the indicated media in the presence of xylose to induce expression of *cdnL* or *cfp* or glucose uninduced control. Cells grown with glucose were pre-depleted of CdnL or CFP for 6 h prior to start of growth curves. Experiments were performed in triplicate and mean is shown. **D.** Phase contrast images of the indicated strains depleted of CdnL or CFP for the indicated number of hours in PYE glucose. Bar = 2 μm. **E.** PCA of cell shape shown in **D**. Scatter plots of normalized values (s.d. from the mean value) for shape modes 1, 2 and 3 which approximately correspond to length, curvature and width. Contours on the left of scatter plots indicate shape of mean and 1 or 2 s.d. from the mean. Mean and SEM included for plots. Statistical analysis performed using one-way ANOVA with Bonferroni's Multiple Comparison Test, n = 424. *** = P < 0.001, ** = P < 0.01, ns = not significant.

of Δ*cdnL* are attributable to loss of CdnL function. To temporally characterize the effects of CdnL depletion on cell morphology, we grew Δ*cdnL* P$_{xylX}$-*cdnL* and Δ*cdnL* P$_{xylX}$-*cfp* in xylose,

washed, grew them in PYE media containing glucose, and imaged them at 0, 7, and 24 hours of growth (Fig 2D). Using CellTool, we performed quantitative shape analysis on phase contrast images of cells producing (time 0) or depleted of CdnL or CFP at the indicated time points. As for Δ*cdnL* (Fig 1C), we found significant differences in the three shape modes reflecting cell length (shape mode 1), curvature (shape mode 2), and width (shape mode 3) (Fig 2E). Specifically, we found that cells become significantly shorter and more curved after 7 hours of CdnL depletion compared to pre-depleted cells. In addition, there was an increase in cell width after 24 hours of CdnL depletion. Collectively, our shape analysis on CdnL-depleted cells indicates that loss of CdnL confers the hypercurvature, decreased length, and increased width of Δ*cdnL* cells. Changes in length and curvature occur more rapidly on CdnL depletion than changes in cell width, suggesting either that these are more direct consequences of CdnL loss and/or that quantifiable changes in width take longer to manifest.

## Genes involved in biosynthetic processes are downregulated in Δ*cdnL*

Having established that the transcriptional regulator CdnL plays a role in regulating growth and morphogenesis, we next sought to understand how it exerts these effects. To identify genes that are misregulated in Δ*cdnL* that may be responsible for the growth and morphology defects that we observe, we extracted mRNA from WT and Δ*cdnL* cells and performed RNA-seq. The results from this analysis showed that approximately 30% of the transcriptome is differentially regulated in cells lacking CdnL (S1 Table).

A closer look at genes that are downregulated in the Δ*cdnL* clone that we used for RNA-seq analysis suggested that a 50 kbp region comprising 51 genes between CCNA_02773 and CCNA_02826 was highly downregulated in Δ*cdnL*. This region is flanked by identical sequences that encode transposases CCNA_02772 and CCNA_02828, suggesting that this region may be deleted in the Δ*cdnL* clone (EG1415) used in our RNA-seq analysis. Indeed, deletion of this region was confirmed by PCR. However, we found no differences in growth or morphology between the Δ*cdnL* clone with the 50 kb deletion (EG1415, Δ*cdnL*Δ50kb) and Δ*cdnL* clones containing this region (e.g. EG1447, used for the analyses in Fig 1) (S2 Fig). Additionally, Δ*cdnL*Δ50kb is a glutamate auxotroph; it behaves similarly to Δ*cdnL* in M2 media supplemented with xylose, glutamate, or glucose; and it can be complemented with expression of *cdnL* from the *xylX* locus (S2 Fig). This suggests that loss of this 50 kb region does not affect the Δ*cdnL* phenotypes we observe. Previously, this region of the chromosome was shown to be readily lost in a background lacking the methyltransferase *ccrM* [20]. Similar to our observations with Δ*cdnL*, loss of this region in Δ*ccrM* did not yield any specific growth advantages.

To ensure that loss of this region did not impact our analysis of the transcriptional consequences specific to loss of CdnL, we analyzed the transcriptome of an independent Δ*cdnL* clone that has the 50 kb region intact (JC784) by microarray analysis (S1 Table). We found significant correlation (Pearson r = 0.6331) between changes in gene expression in Δ*cdnL* as compared to WT in the RNA-seq dataset using EG1415 and the microarray dataset using JC784 (Fig 3A, S3 Fig). Overlap in genes differentially regulated in the two datasets was found to be significant using Fisher's exact test (p-value = 2.2e$^{-16}$ and odds ratio = 5.32). We generally used the intersection of our two datasets (RNA-seq of EG1415 and microarray of JC874) as a very conservative and rigorous representation of the consequences of deleting *cdnL* on the *Caulobacter* transcriptome. In total, we found 247 genes at least 2-fold upregulated in Δ*cdnL* and 278 genes at least 2-fold downregulated in Δ*cdnL* out of the 3691 transcripts detected in both datasets (S3 Fig).

To obtain an overview of how loss of CdnL affects the transcriptome we used DAVID [21] functional annotation analysis to functionally categorize genes that had at least a two-fold

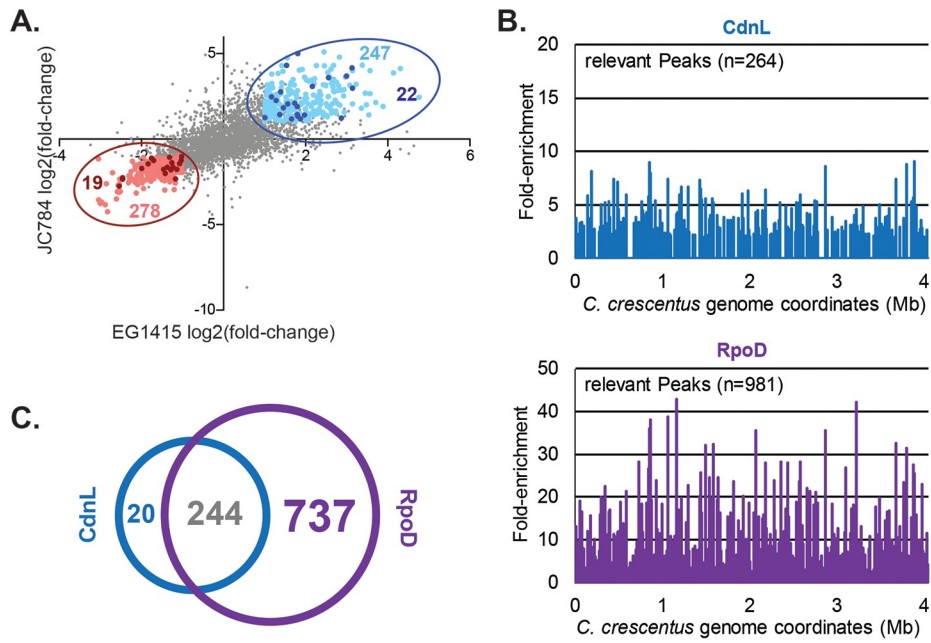

**Fig 3. Transcriptional changes in Δ*cdnL* are broad and mostly indirect. A.** Plot comparing the log₂(fold-change) of Δ*cdnL* compared to WT for each transcript in the two transcriptomics datasets: microarray of JC784 and RNA-seq of EG1415. Pearson's correlation = 0.6331. The 278 genes at least 2-fold downregulated in both datasets are highlighted in pink and the 247 genes at least 2-fold upregulated in both are highlighted in light blue. Putative direct targets of CdnL that are at least 2-fold altered in Δ*cdnL* in both datasets are highlighted in red and dark blue (listed in S3 Table). **B.** Peaks representing fold-enrichment of loci across the genome immunoprecipitated using CdnL (blue) or RpoD (purple) antibodies. **C.** Intersection of the peaks enriched at least 2-fold with CdnL or RpoD antibodies by ChIP-seq.

change in transcript abundance in Δ*cdnL* in both the RNA-seq and microarray datasets. Of the genes that are downregulated in Δ*cdnL*, we found an overrepresentation of genes involved in biosynthetic and bioenergetic pathways including amino acid, nucleotide, fatty acid, lipid, cell wall, cell membrane, and central carbon metabolism (S2 Table). Genes that are over 2-fold upregulated in Δ*cdnL* clustered into categories involved in transcription, signal transduction, transport, and motility.

## Identification of the CdnL direct regulon

Given the large number of changes in the transcriptome that occur in Δ*cdnL*, we sought to identify which genes were likely direct targets of CdnL. To this end, we identified sites of CdnL occupancy on the chromosome by chromatin immunoprecipitation using CdnL antisera followed by sequencing (ChIP-seq) and identified 264 peaks with at least a 2-fold enrichment (Fig 3B, S3 Table). These peaks overlapped with putative regulatory regions of 406 genes (roughly 10% of those in the genome), representing the potential direct targets of CdnL. We performed DAVID functional annotation analysis on the 406 CdnL-associated genes we identified and found that they are particularly enriched for protein components of the ribosome, rRNAs, and aminoacyl-tRNA biosynthesis (S4 Table). Additional enriched functional groups include genes involved in transcriptional regulation, nucleotide metabolism, energy metabolism, chaperones, and proteases. Since CdnL has been previously shown to bind to promoters associated with σ⁷⁰ (RpoD) [3], we performed ChIP-seq with RpoD antisera and identified 981 peaks with at least a 2-fold enrichment (Fig 3B, S3 Table). Comparison of CdnL and RpoD ChIP-seq datasets revealed that CdnL and RpoD co-localize at 244 sites and that only 20 out of

the 264 CdnL peaks are not also associated with RpoD (Fig 3C), suggesting that CdnL is associated with RpoD-containing RNAP holoenzyme.

Having identified the sites of CdnL occupancy across the *Caulobacter* genome, we determined the putative direct regulon of CdnL by comparing our ChIP-seq data to our transcriptomics datasets (Fig 3A). Out of the 247 genes that are upregulated and 278 genes that are downregulated at least 2-fold in both EG1415 and JC784, only 22 and 19 of the corresponding promoter regions immunoprecipitate with CdnL, respectively (Fig 3A, S3 Table). We note that our definition of direct targets is particularly stringent since we require a gene to be differentially regulated at least 2-fold in both transcriptomics datasets. Moreover, not all of the promoters associated with CdnL by ChIP-seq control transcripts that were detected in both transcriptomics datasets, and our analysis does not take into consideration putative operon structure. It is therefore a conservative view of the direct CdnL regulon. Nevertheless, the weak overlap between the ChIP-seq and transcriptomics datasets suggests that most of the changes that occur in the transcriptome in Δ*cdnL* may be indirect transcriptional or post-transcriptional changes. This is perhaps not surprising since direct targets of CdnL include ribosomal components, transcriptional regulators, and small RNAs which might collectively enable a cascade of indirect gene regulation. In addition, many of the transcriptional changes we observe in Δ*cdnL* may reflect long term adaptation to metabolic and other stresses imposed by CdnL loss.

## Δ*cdnL* cells have reduced levels of central carbon and TCA metabolites required for synthesizing macromolecules critical for growth

Since Δ*cdnL* cells exhibit growth defects and our transcriptome analysis revealed downregulation of pathways involved in macromolecular biosynthesis, we hypothesized that Δ*cdnL* cells may have limited amounts of substrates available for proliferative processes. To identify how the metabolome changes when *cdnL* is deleted, we extracted metabolites from WT and Δ*cdnL* cells grown in PYE, M2GG, M2G (grown in M2GG, washed and grown in M2G for 12 hours) or HIGG media or from CdnL-depleted cells grown in PYE, and performed LC/MS analysis of polar metabolites. Consistent with what we inferred from our transcriptomic data, we found glycolytic and tricarboxylic acid (TCA) intermediates, amino acids, nucleotides, and their derivatives to be significantly altered, suggesting that central carbon metabolism and carbohydrate utilization is reduced in cells lacking CdnL (Fig 4, S5 Table, S4 Fig). Specifically, we found biosynthetic precursors and cofactors such as pyruvate, $NAD^+$, ATP, and UDP-N-acetyl glucosamine are significantly reduced in Δ*cdnL* cells while D-xylose, uric acid, the TCA intermediates fumarate and malate, the pyrimidine and amino acid precursor dihydroorotate, the leucine synthesis intermediate 2-isopropylmalic acid, and the storage molecule 3-hydroxybutyric acid are significantly higher in Δ*cdnL* across all conditions compared to WT (Fig 4, S5 Table, S4 Fig).

Since we performed our transcriptomics analysis in PYE, we compared our RNA-seq data to our PYE metabolome dataset. In *Caulobacter*, glucose is metabolized to yield substrates that feed into the TCA cycle via the Entner-Doudoroff pathway [22,23]. Since intermediates in and products of this glycolytic pathway are reduced in Δ*cdnL*, we looked at our transcriptomic data to see if the levels of transcripts encoding any of the enzymes in this pathway are changed. We found that *CCNA_01562* and *CCNA_01560* are significantly downregulated suggesting that disruption in flux through the Entner-Doudoroff pathway may, indeed, lead to low levels of pyruvate and phosphoenolpyruvate (PEP) (Fig 4B, S1 Table). *Caulobacter* can synthesize all amino acids *de novo* using TCA cycle intermediates [22]. During growth in media without amino acids, *Caulobacter* cells can convert PEP to oxaloacetate to replenish TCA intermediates

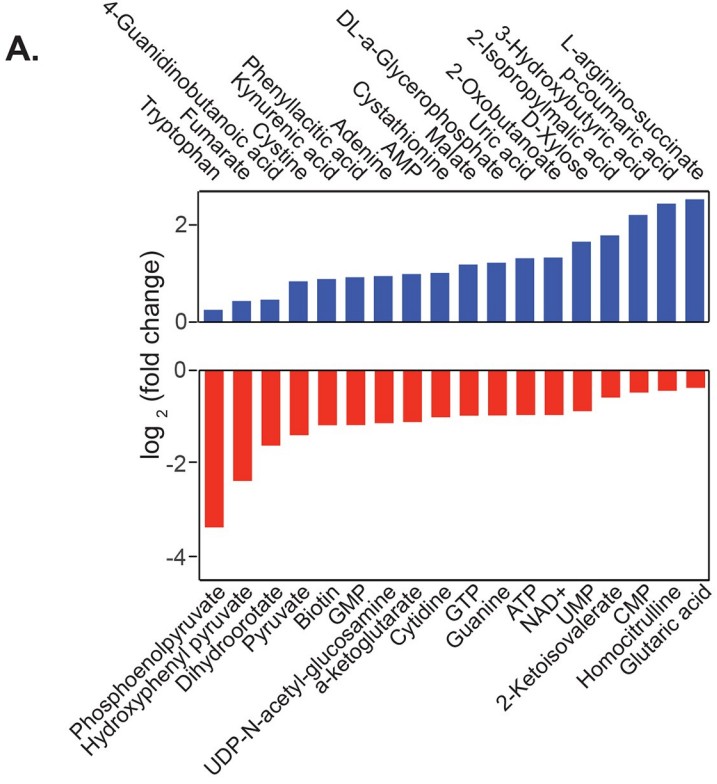

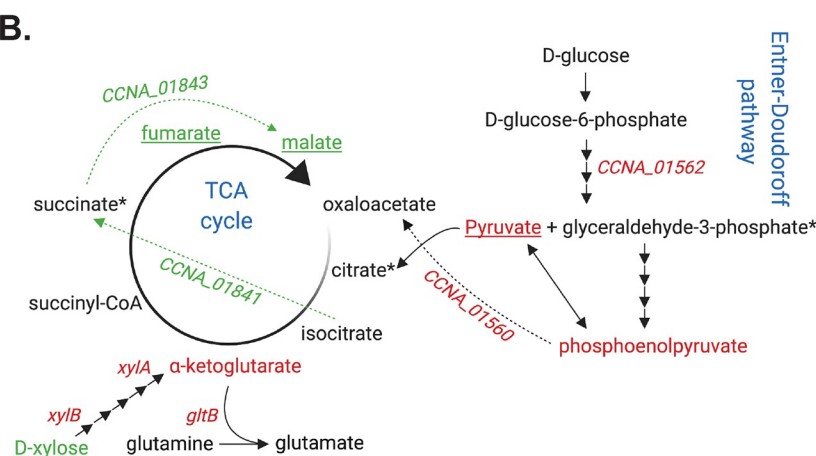

**Fig 4. CdnL is required for metabolic homeostasis in *Caulobacter*. A.** Graph showing significantly altered metabolites (P < 0.05) in Δ*cdnL* compared to WT grown in PYE from S5 Table. **B.** Summary figure depicting changes in glucose metabolism and TCA cycle in Δ*cdnL* cells as compared to WT. Genes and metabolites shown in red are downregulated and in green are upregulated. Arrows in green indicate glyoxylate bypass pathway. Underlined metabolites are changed across all media types, whereas those in black are unchanged. Dashed arrows indicate pathways that are active only under some conditions. * indicates metabolites not detected. n = 3 for each strain in each condition.

[22]. We found that *CCNA_01560*, which is predicted to convert PEP to oxaloacetate, is also highly downregulated in Δ*cdnL*. Thus, our observations of low α-ketoglutarate levels and glutamate auxotrophy may arise due to reduced flux of pyruvate and PEP into the TCA cycle (Fig 4).

Interestingly, D-xylose levels were elevated in Δ*cdnL* cells in all media tested, although xylose was not provided exogenously. Conversely, genes required to metabolize xylose are downregulated in Δ*cdnL*. Previously, xylose accumulation without an ability to metabolize xylose has been shown to upregulate isocitrate lyase (*CCNA_01841*) and initiate the glyoxylate bypass which promotes conversion of isocitrate to succinate, bypassing key TCA cycle intermediates such as α-ketoglutarate [22,24]. Additionally, upregulation of malate synthase (*CCNA_01843*) combines glyoxylate produced by isocitrate lyase to acetyl-coenzyme A to produce malate, allowing a modified TCA cycle to continue [22]. Consistently, we find that *CCNA_01841* and *CCNA_01843* are over four-fold and two-fold upregulated in Δ*cdnL*, respectively (S1 Table), suggesting low levels of α-ketoglutarate may arise due to activation of the glyoxyate bypass in addition to low levels of pyruvate and PEP. Since α-ketoglutarate is used to synthesize glutamate (Fig 4B), the glutamate auxotrophy of Δ*cdnL* cells may arise due to low amounts of α-ketoglutarate produced by the TCA cycle. Additionally, we found that *gltB*, which is essential for glutamate biosynthesis from α-ketoglutarate and glutamine, is over 4-fold downregulated in our transcriptomic analysis (Fig 5A, S1 Table). Collectively, our data indicate that changes in the transcriptome that arise when *cdnL* is deleted have detrimental consequences on metabolic pathways disrupting levels of key metabolites required for energy production and for amino acid and nucleotide biosynthesis.

## Δ*cdnL* cells have low levels of the cell wall precursor and changes in cell wall crosslinking

Our metabolomics analysis showed that UDP-N-acetylglucosamine and PEP, substrates required for synthesis of the PG precursor lipid II, are significantly lower in Δ*cdnL* compared to WT (Fig 4). Additionally, our transcriptomic analysis indicated that a number genes required for lipid II biosynthesis are highly downregulated (Fig 5A) leading us to postulate that PG material maybe limiting in Δ*cdnL* cells. Since PG synthesis is required for both cell shape maintenance and cell envelope integrity, changes in abundance of lipid II might, at least partially, underlie the shape and cell lysis phenotypes observed for Δ*cdnL* cells. We directly compared lipid II levels in WT and Δ*cdnL* cells and found that Δ*cdnL* cells have a striking 45% reduction in lipid II levels as compared to WT (Fig 5B). Previously, work in *Vibrio cholerae* showed that low levels of lipid II contribute to a decrease in overall PG density [25,26]. Consistent with low levels of the lipid II substrate for PG polymerization, Δ*cdnL* cells have 23% less PG polymer than WT (Fig 5C).

To assess if CdnL-mediated changes in transcription result in alterations to PG metabolic activities in addition to lipid II synthesis, we performed muropeptide analysis on PG isolated from WT and Δ*cdnL* cells to identify changes in PG chemistry. Although overall PG crosslinking is unaffected, Δ*cdnL* cells had significantly more dimers and fewer trimers (higher order crosslinks) than WT (Fig 5D, S5 Fig). The mutant also shows an increased chain length, as the relative amount of anhydro muropeptides (glycan chain termini) were significantly reduced in Δ*cdnL* cells (Fig 5D, S5 Fig). Moreover, Δ*cdnL* cells had more pentapeptides and glycine-containing muropeptides compared to WT (Fig 5D, S5 Fig). Thus, the shape and envelope integrity defects observed in Δ*cdnL* cells may arise, at least in part, from a combination of low levels of lipid II and significant changes in the chemical structure of the cell wall.

## Pathways impacting PG metabolism become essential in Δ*cdnL*

To gain insight into which transcriptional and metabolic changes in Δ*cdnL* may be most relevant to fitness, we performed comparative transposon sequencing (Tn-seq) on WT and Δ*cdnL* strains. Putative negative genetic interactions were detected between Δ*cdnL* and genes

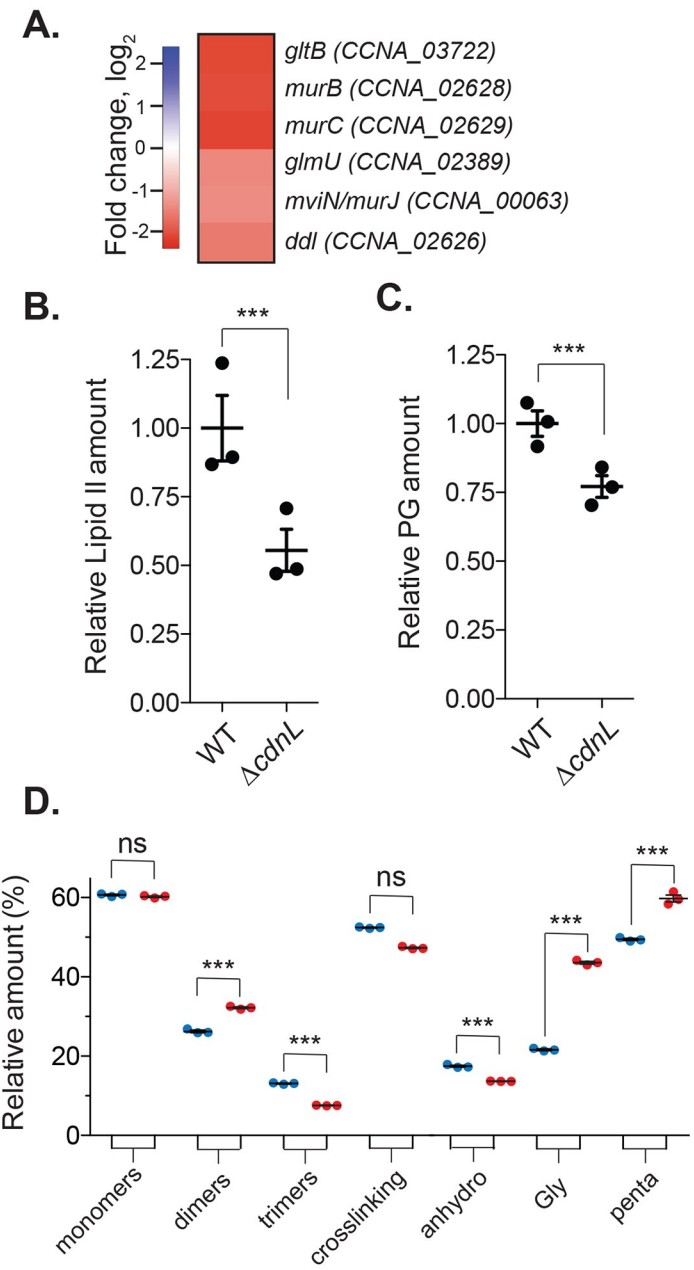

**Fig 5. CdnL impacts cell wall precursor synthesis and cell wall metabolism. A.** Heatmap showing changes of the indicated transcripts involved in lipid II synthesis and cell wall remodeling in WT (EG865) and Δ*cdnL* (EG1415) using RNA-seq. **B-C.** Relative amount of lipid II and peptidoglycan in WT (EG865) and Δ*cdnL* (EG1447) cells. **D.** Indicated muropeptide species from WT (EG865) and Δ*cdnL* (EG1447). X-axis shown in **D** are (1–6 anhydro) N-acetyl muramic acid (anhydro), Gly containing muropeptides (Gly), and pentapeptides (penta). Values in **B-D** are from three independent cultures and mean and SEM are indicated. Statistical analysis performed using unpaired t test. *** = P < 0.001, ** = P < 0.01, * = P < 0.05.

involved in carbon metabolism, energy production, signaling, transport, and transcriptional regulation (S4 Table). Consistent with the morphological defects and low amounts of lipid II observed for Δ*cdnL* cells, we found several normally non-essential genes involved in pathways that interact with PG synthesis could not be disrupted in the Δ*cdnL* background (Fig 6A, S6

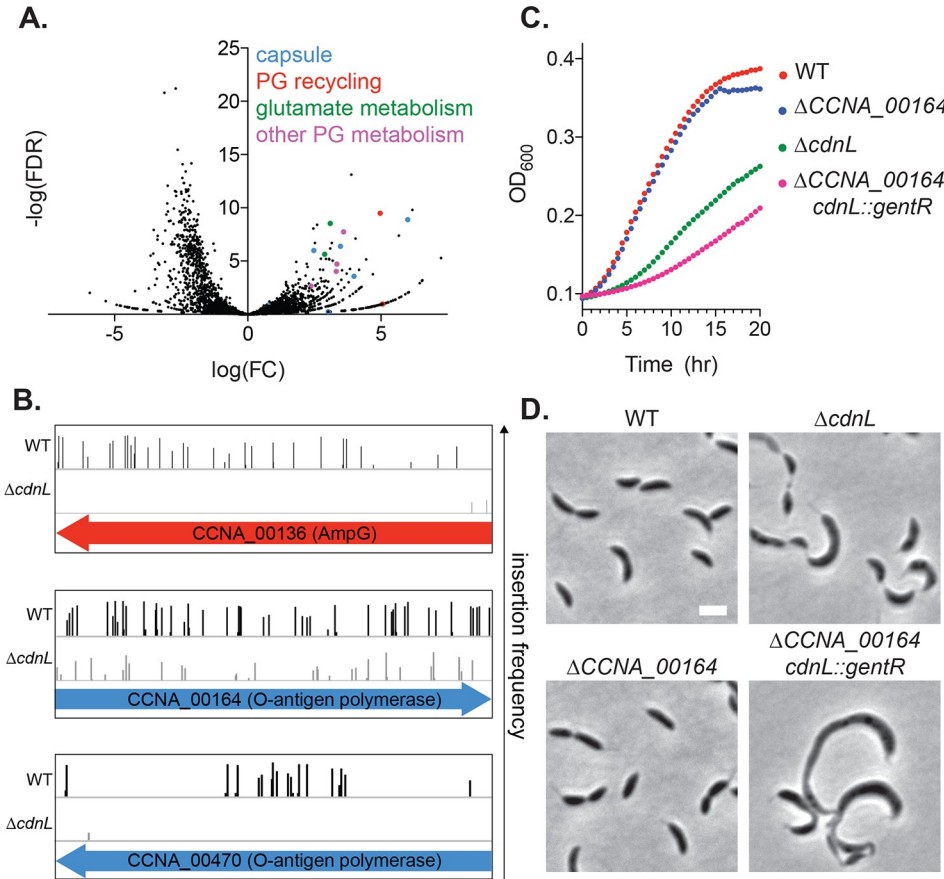

**Fig 6. Mutations in cell wall recycling and capsule biosynthetic genes are synthetic lethal with ΔcdnL. A.** Volcano plot representation of Tn-seq analysis. Negative $\log_{10}$ of the false discovery rate (-log(FDR)) is plotted against $\log_2$ of the fold change in number of unique transposon insertions in each gene in WT (EG865) vs *ΔcdnL* (EG1447). Genes that become essential in the *ΔcdnL* background and are discussed in this study are color-coded. **B.** Transposon insertion profile for selected genes from **A. C-D.** Growth curves and phase contrast images for WT (EG865), *ΔcdnL* (EG1447), *ΔCCNA_00164* (EG2604) and *ΔcdnL ΔCCNA_00164* (EG2804). Bar = 2 μm.

Table). Specifically, we identified genes involved in glutamate metabolism, PG metabolism, PG recycling, and capsule synthesis in which we recovered far fewer transposon insertions in the *ΔcdnL* background than in WT (Fig 6A and 6B, S6 Table). Of the genes involved in glutamate metabolism, GdhZ and MurI are predicted to become essential in *ΔcdnL*. Each of these proteins link glutamate metabolism to morphogenesis: MurI by making D-Glu for lipid II synthesis and GdhZ via its effects on the morphogenetic protein FtsZ [27,28].

Our Tn-seq analysis also indicated that *ampG (CCNA_00136)* and *ampD (CCNA_02650)*, genes required for PG recycling, become essential in *ΔcdnL* (Fig 6B, S6 Table). AmpG is a permease that allows transport of cell wall material from the periplasm to the cytoplasm and AmpD degrades transported cell wall material to provide building blocks for lipid II synthesis [29]. We attempted to delete *cdnL* in a *ΔampG* background, but were unable to obtain clones with deletions in both *ampG* and *cdnL*, supporting the genetic interaction suggested from the Tn-seq. Since lipid II levels are low in *ΔcdnL*, the cell apparently relies on pathways that allow for recycling of the existing cell wall material for survival.

In addition to PG metabolic genes, we found putative synthetic lethal interactions between capsule (exopolysaccharide) biosynthetic genes and *cdnL* (Fig 6A,B, S6 Table). During capsule

synthesis, saccharide precursors are synthesized on the lipid anchor undecaprenyl phosphate (Und-P) in the cytoplasm, flipped to the periplasmic space, and assembled into a polysaccharide that is exported to the outer surface of the cell [30]. Thus, PG synthesis and capsule synthesis each use Und-P as a lipid carrier for synthesis and export into the periplasmic space. Previous work in *Escherichia coli* and *Bacillus subtilis* has shown that disruption of pathways that use Und-P can lead to Und-P sequestration and cause severe growth and morphological defects by inhibiting lipid II synthesis [31–33]. Similarly, we propose that disrupting the capsule biosynthesis pathway in *Caulobacter* may sequester Und-P in dead-end intermediates, limiting lipid II synthesis. We attempted to delete each of the capsule genes predicted by Tn-seq to genetically interact with *cdnL* in Δ*cdnL* cells. We were unsuccessful with the exception of *CCNA_00164*, which encodes a putative O-antigen polymerase ligase and which had the weakest predicted genetic interaction with *cdnL* of the capsule genes (Fig 6A and 6B and S6 Table). Nevertheless, deletion of both *cdnL* and *CCNA_00164* resulted in cells with morphological defects that are more severe than Δ*cdnL* alone (Fig 6D). Moreover, these double mutant cells grew more slowly than Δ*cdnL* (Fig 6C). Deletion of *CCNA_00164* by itself had no obvious effect on growth or morphology. Our data support the model that mutations in the capsule biosynthesis pathway sequester Und-P and limit its availability for lipid II synthesis, exacerbating the existing lipid II deficiency of Δ*cdnL* cells to a lethal level. Collectively, our transcriptomics, biochemical analyses, and genetic interaction results indicate that CdnL is required to maintain adequate availability of the PG precursor lipid II to support normal growth and morphology.

## Discussion

In this work, we unveil a role for the conserved, global transcriptional regulator CdnL in maintaining transcription of biosynthetic genes required for metabolic homeostasis in *Caulobacter*. We show that in Δ*cdnL* cells, transcripts of many classes of biosynthetic genes are downregulated, with physiologically critical defects in glutamate and lipid II metabolism. These changes in metabolic pools affect growth and morphology at least in part by limiting macromolecules required for cell growth and envelope expansion. Downregulation of genes involved in biosynthetic and bioenergetics pathways in Δ*cdnL* cells, an altered metabolic landscape in Δ*cdnL*, and the reliance of Δ*cdnL* cells on exogenous glutamate, which is a central node in multiple biosynthetic pathways, implicate CdnL directly and indirectly in expression of genes required for central metabolism and macromolecular synthesis [34,35].

Previously, CdnL has been shown to directly bind to RNAP and stabilize the open promoter complex required for transcription of rRNA promoters in *Mycobacteria* and *M. xanthus* [3,5]. Similarly, *Caulobacter* CdnL has been shown to bind RNAP and regulate transcription from an rRNA promoter [11]. Here, we demonstrate that loss of CdnL has global effects on various aspects of bacterial metabolism and CdnL directly associates with hundreds of loci implicated in biosynthetic and regulatory pathways. Interestingly, however, transcriptomic changes that occur during loss of CdnL in *Caulobacter* are the opposite of what is observed in *Mycobacterium smegmatis*. CarD depletion in *M. smegmatis* leads to upregulation of genes encoding the translation machinery and enzymes in metabolic pathways such as amino acid biosynthesis and to downregulation of genes involved in degradation of amino acids, fatty acid metabolism and membrane proteins [2]. Although loss of CdnL generally has opposite effects on expression of metabolic genes in these two divergent bacteria, CdnL-mediated effects on the transcriptome impact proliferative and metabolic processes and are thereby essential for proper growth in *Caulobacter* and viability in *Mycobacteria*. These differences may be in part attributable to immediate (i.e. CarD depletion) versus long term (i.e. Δ*cdnL*) effects of CdnL loss on the transcriptome.

In *Caulobacter*, loss of CdnL causes aberrant morphogenesis, at least in part, by limiting production of lipid II. Low levels of lipid II are expected to lead to slowed PG synthesis and a decrease in PG density, and could impact crosslinking by tempering the activity of class A PBPs [36,37]. These enzymes polymerize glycan strands via transglycosylation reactions and crosslink peptide stems through transpeptidation reactions [36]. The transpeptidation activity of class A PBPs requires ongoing glycan strand polymerization [36,37]. Thus, conditions that lower lipid II levels, such as loss of CdnL, are expected to lead to a decrease in PG crosslinking. Consistently, we observe a decrease in trimers, but total crosslinking is not affected. Interestingly, we also observed an increase in glycine-containing muropeptides, with a D-Ala-Gly terminal in the peptide stem instead D-Ala-D-Ala. We did not detect lipid II containing the D-Ala-Gly peptide stem during our precursor analysis suggesting that accumulation of D-Ala-Gly muropepetides in the cell wall may occur via PBPs after PG polymerization. In *Caulobacter*, the accumulation of glycine in PG does not affect cell growth and morphogenesis, so we do not think the morphogenetic defects of Δ*cdnL* are attributable to this change in PG chemistry [38].

The effects of *cdnL* deletion are in some ways reminiscent of loss of the master regulator and RNA chaperone Hfq in *Caulobacter* [39]. Deletion of *hfq* alters the levels of several metabolites, including glutamate and components of the TCA cycle. Importantly, this includes accumulation of α-ketoglutarate which, in turn, causes reduced synthesis of lipid II and pleiotropic cell shape defects [39]. Though both Hfq and CdnL are required to maintain sufficiently high levels of lipid II to support growth and stereotyped cell shape, Δ*cdnL* and Δ*hfq* have distinct shape phenotypes, likely through differential impacts on the transcriptome and metabolome beyond lipid II biosynthetic pathways.

We first identified *Caulobacter* CdnL as a loss of function suppressor of the dominant lethal cell wall defects associated with producing the cell division protein FtsZ lacking its disorder C-terminal linker (FtsZΔCTL). Strangely, FtsZΔCTL production and loss of CdnL lead to defects in cell wall metabolism causing inevitable or occasional cell lysis, respectively. Each also causes a decrease in higher order PG crosslinking and increased length of glycan strands. Why then does Δ*cdnL* suppress rather than exacerbate the ΔCTL-induced PG metabolic defects? We can think of two likely possibilities. The first is that limiting the availability of lipid II substrate in FtsZΔCTL-producing cells slows PG synthesis, and this prevents the toxic misregulation of PG metabolism induced by FtsZΔCTL. Alternatively, the sweeping metabolic changes and consequent effects on growth and cell wall metabolism in Δ*cdnL* suppresses the effects of ΔCTL through a combination of PG metabolic changes. As we do not yet know what specific defects on PG metabolism are imposed by ΔCTL, we cannot distinguish between these two (or other) possibilities.

Depletion of CdnL in *M. xanthus* results in cell filamentation, consistent with a conserved impact on morphogenesis [6]. The effects, if any, of CarD depletion in *Mycobacteria* on cell morphology or cell wall metabolism have not been reported, nor have the effects of CdnL depletion on the *M. xanthus* transcriptome or metabolome. Future work in diverse organisms will reveal if the role of CdnL as a global regulator of both morphogenesis and metabolism is conserved across bacteria.

## Methods

### Bacterial strains, plasmids, growth conditions

*Caulobacter crescentus* NA1000 strains were grown in peptone yeast extract medium (PYE), M2G [40], M2GG (M2G with 0.15% sodium glutamate), or Hutner base imidazole-buffered glucose glutamate medium (HIGG) [41]. Xylose was used at 0.3% for induction experiments.

EG1415 and JC874 were used for RNA-seq and microarray experiments, respectively. All other experiments with Δ*cdnL* were performed using EG1447 (clean, in-frame deletion of *cdnL*) or derivatives thereof. JC874, EG1447, and EG1416, and each contain the 50 kb region that is deleted in EG1415. Growth curves were performed in a Tecan Infinite M200 Pro plate reader, 100 μL culture volume, with each strain monitored in biological triplicate at $OD_{600}$ every 30 min with intermittent shaking just prior to each $OD_{600}$ reading. Note that under these growth conditions, WT *Caulobacter* doubles in ~5 hours in both PYE and M2G media. CdnL depletion growth curves were performed by growing cells media containing 0.3% xylose, washed three times and resuspended in media containing 0.3% glucose. Cells were pre-depleted of CdnL (or CFP) for 6 h prior to starting the growth curves. Antibiotics for growing *Caulobacter* and cloning and purifying plasmids from *E. coli* were used at concentrations in liquid (solid) media as described in Woldemeskel *et al* 2017 [42]. *C. crescentus*: kanamycin 5 (25) μg/mL, gentamicin 1 (5) μg/mL. *E. coli*: ampicillin 50 (100) μg/mL, kanamycin 30 (50) μg/mL, gentamicin 15 (20) μg/mL, and spectinomycin 50 (50) μg/mL.

## Phase contrast, ensemble fluorescence microscopy, and image analysis

Cells in exponential phase of growth were spotted on 1% agarose pads and imaged using a Nikon Eclipse Ti inverted microscope equipped with a Nikon Plan Fluor 100X (NA1.30) oil Ph3 objective and Photometrics CoolSNAP HQ$^2$ cooled CCD camera. Chroma filter cube ET-dsRED was used for mCherry. Images were processed using Adobe Photoshop. Principal component analysis to identify shape variations were performed using CellTool [16] after phase contrast images were converted to binary masks in ImageJ [43] and edited in Adobe Photoshop to remove overlapping cells, out of focus cells, and cells not completely in the field of view.

## Immunoblotting to assess CdnL levels

His$_6$-SUMO-CdnL was overproduced in Rosetta (DE3) pLysS *E.coli* cells from pEG1129 by inducing for 3 hr at 30 °C with 0.5 mM IPTG at $OD_{600}$ of 0.5. Cells were harvested by centrifugation, resuspended in lysis buffer (50 mM Tris-HCl, pH 8.0, 300 mM KCl, 20 mM imidazole, and 10% glycerol), lysed with 1 mg/mL lysozyme, sonicated and centrifuged at 4 °C for 30 min at 15000 x g. Supernatant was loaded onto a HisTrap FF 1 mL column (GE Life Sciences) and eluted with lysis buffer with 300 mM imidazole, cleaved with His$_6$-Ulp1 (SUMO protease) at 1:500 (protease: fusion) molar ratio overnight while dialyzing into lysis buffer. Sample was reloaded onto HisTrap FF 1mL and flow through was collected to separate His$_6$-SUMO and His$_6$-Ulp1 from CdnL. CdnL fraction was dialyzed into PBS prior to antibody production.

Cells in log phase were isolated and lysed in SDS-PAGE loading buffer by boiling for 5 min. Standard procedures were used for SDS-PAGE and transfer to nitrocellulose membrane. CdnL antisera was generated by immunizing a rabbit with CdnL purified as above (Pocono Rabbit Farm & Laboratory). Specificity was determined using cell lysates with deleted or over-expressed *cdnL*. CdnL antisera was used at 1:5000.

## RNA-seq preparation

Cultures of three independent colonies of WT (EG865) and Δ*cdnL* (EG1415, Δ50kb) cells were grown in PYE and harvested at $OD_{600}$ of 0.45 and total RNA was extracted using PureLink RNA Mini Kit from Thermo Fisher using the protocol provided with the kit. Briefly, cells were harvested, lysed, and homogenized using lysozyme, SDS, provided lysis buffer, and homogenizer. Nucleic acids were extracted with ethanol and loaded onto the provided spin cartridge. DNA-free total RNA was extracted using the on-column PureLink DNAse Treatment

protocol. PureLink DNAse mixture was directly added on to the spin cartridge membrane, incubated for 15 min and washed using provided wash buffers and RNA was eluted with RNAse-free water. Finally, RNA quality was quantified using a bioanalyzer and rRNA was removed using the Ribo-Zero rRNA Removal Kit (Gram-Negative Bacteria). RNA-seq libraries were prepared using the Illumina TruSeq stranded RNA kit and sequenced on an Illumina HiSeq 2500. Data analysis was performed using the Illumina's CASAVA 1.8.2, Bowtie2 (v 2.2.5), and Bioconductor's DESeq package. rRNA clean up, quality control analysis, library prep, sequencing and analysis was performed by the Johns Hopkins University School of Medicine Next Generation Sequencing Center. RNA-seq data have been deposited in the Sequence Read Archive (SRA) under accession numbers SRX7083467-SRX7083472 and BioProject number PRJNA586876.

## Microarray preparations

Biological triplicates of WT (JC450) and Δ*cdnL* (JC784) cultures grown in PYE were harvested at $OD_{660}$ of 0.3. RNA was extracted with Trizol, purified using Ambion PureLink RNA mini-kit according to manufacturer's protocol. Double stranded cDNA was prepared with Roche Double Stranded cDNA synthesis kit. Microarray analysis was performed on Nimblegen custom chips following the manufacturer's protocol for labeling, hybridization on 4-plex chips, and washing. Scanning was performed with Agilent microarray scanner at 3 μm and normalization was performed with Nimblescan implemented RMA algorithm (3 x 4-plex chips in total, all samples from the same culture conditions). The statistical analysis of significance was performed with R using the Significance of Analysis of Microarrays (SAM) method implemented in the library *siggenes* [44]. Microarray data have been deposited to the Gene Expression Omnibus (GEO) database under accession number GSE139873.

## DAVID analysis and functional clustering

Genes with a two-fold change in transcript levels in both RNA-seq and microarray analysis, genes significantly enriched by ChIP-seq using CdnL antisera, and genes with negative genetic interactions with Δ*cdnL* were functionally categorized using DAVID [21] with a medium classification stringency and default parameters.

## Chromatin Immunoprecipitation coupled to deep sequencing (ChIP-seq)

Cultures of exponentially growing ($OD_{660nm}$ of 0.5, 80 ml culture in PYE per sample) *C. crescentus* NA1000 WT strain were supplemented with 10 μM sodium phosphate buffer (pH 7.6) and then treated with formaldehyde (1% final concentration) at RT for 10 min to achieve crosslinking. Subsequently, the cultures were incubated for an additional 30 min on ice and washed three times in phosphate buffered saline (PBS, pH 7.4). The resulting cell pellets were stored at -80°C. After resuspension of the cells in TES buffer (10 mM Tris-HCl pH 7.5, 1 mM EDTA, 100 mM NaCl) containing 10 mM of DTT, the cell resuspensions were incubated in the presence of Ready-Lyse lysozyme solution (Epicentre, Madison, WI) for 10 minutes at 37°C, according to the manufacturer's instructions. Lysates were sonicated (Bioruptor Pico) at 4°C using 15 bursts of 30 sec to shear DNA fragments to an average length of 0.3–0.5 kbp and cleared by centrifugation at 14,000 rpm for 2 min at 4°C. The volume of the lysates was then adjusted (relative to the protein concentration) to 1 ml using ChIP buffer (0.01% SDS, 1.1% Triton X-84 100, 1.2 mM EDTA, 16.7 mM Tris-HCl [pH 8.1], 167 mM NaCl) containing protease inhibitors (Roche) and pre-cleared with 80 μl of Protein-A agarose (Roche, www.roche.com) and 100 μg BSA. Five percent of the pre-cleared lysates were kept as total input samples (negative control samples). The rest of the pre-cleared lysates was then incubated overnight at

4˚C with polyclonal rabbit antibodies targeting CdnL (1:1,000 dilution) or monoclonal mouse antibodies targeting RpoD/Sigma70 (1:250 dilution; Neoclone #WP004), respectively. The immuno-complexes were captured after incubation with Protein-A (CdnL) or Protein-G (RpoD/Sigma70) agarose beads (pre-saturated with BSA) during a 2 h incubation at 4˚C and then, washed subsequently with low salt washing buffer (0.1% SDS, 1% Triton X-100, 2 mM EDTA, 20 mM Tris-HCl pH 8.1, 150 mM NaCl), with high salt washing buffer (0.1% SDS, 1% Triton X-100, 2 mM EDTA, 20 mM Tris-HCl pH 8.1, 500 mM NaCl), with LiCl washing buffer (0.25 M LiCl, 1% NP-40, 1% deoxycholate, 1 mM EDTA, 10 mM Tris-HCl pH 8.1) and finally twice with TE buffer (10 mM Tris-HCl pH 8.1, 1 mM EDTA). The immuno-complexes were eluted from the Protein-A or Protein-G agarose beads with two times 250 μL elution buffer (SDS 1%, 0.1 M NaHCO3, freshly prepared) and then, just like the total input samples, incubated overnight with 300 mM NaCl at 65˚C to reverse the crosslinks. The samples were then treated with 2 μg of Proteinase K for 2 h at 45˚C in 40 mM EDTA and 40 mM Tris-HCl (pH 6.5). DNA was extracted using phenol:chloroform:isoamyl alcohol (25:24:1), ethanol-precipitated using 20 μg of glycogen as a carrier and resuspended in 50 μl of DNAse/RNAse free water.

Immunoprecipitated chromatin was used to prepare sample libraries used for deep-sequencing at Fasteris SA (Geneva, Switzerland). ChIP-seq libraries were prepared using the DNA Sample Prep Kit (Illumina) according to the manufacturer's instructions. Single-end runs (50 cycles) were performed on an Illumina HiSeq2500 instrument, yielding several million reads. The single-end sequence reads (stored as FastQ files) were mapped against the genome of *C. crescentus* NA1000 (NC_011916.1) using Bowtie version 0.12.9 (-qS -m 1 parameters, http://bowtie-bio.sourceforge.net/). ChIP-seq read sequencing and alignment statistics are summarized in S3 Table. The standard genomic position format files (BAM, using Samtools, http://samtools.sourceforge.net/) were imported into SeqMonk version 1.45.4 (Braham, http://www.bioinformatics.babraham.ac.uk/projects/seqmonk/) to build ChIP-seq normalized sequence read profiles. Briefly, the genome was subdivided into 50 bp probes, and for every probe, we calculated the number of reads per probe as a function of the total number of reads (per million, using the Read Count Quantitation option). Analysed data as shown in Fig 3 are provided in S3 Table).

Using the web-based analysis platform Galaxy (https://usegalaxy.org), CdnL and RpoD/Sigma70 ChIP-seq peaks were called using MACS2 callpeak software (Galaxy Version 2.1.1.20160309.6) relative to their total input DNA samples. The q-value (false discovery rate, FDR) cut-off for called peaks was 0.05. CdnL and RpoD/Sigma70 peaks were rank-ordered according to fold-enrichment (S3 Table), and peaks with a fold-enrichment values >2 were retained for further analysis. CdnL statistical peaks were annotated using SeqMonk (Surrounding CDS option). Sequence data have been deposited to the Gene Expression Omnibus (GEO) database under accession GSE140134 and samples GSM4154751–GSM4154754.

## Transposon library preparation, sequencing, and analysis

Two Tn-seq libraries each were generated for WT (EG865) and Δ*cdnL* (EG1447). 1L PYE cultures were harvested at OD$_{600}$ of 0.4–0.6, washed 5 times with 10% glycerol, and electroporated with the Ez-Tn5 <Kan-2> transposome (Epicentre). Cells were recovered for 90 minutes at 30 °C with shaking, and plated on PYE-Kan plates. WT libraries were grown for 5 and 6 days and Δ*cdnL* libraries were grown for 6 and 10 days. Colonies were scraped off plates, combined, resuspended to form a homogeneous solution in PYE, and flash frozen in 20% glycerol. Genomic DNA for each library was extracted from an aliquot using the DNeasy Blood and Tissue Kit (Qiagen). Libraries were then prepared for Illumina Next-Generation sequencing through

sequential PCR reactions. The first PCR round used arbitrary hexamer primers with a Tn5 specific primer going outward. The second round used indexing primers with unique identifiers to filter artifacts arising from PCR duplicates. Indexed libraries were pooled and sequenced at the University of Massachusetts Amherst Genomics Core Facility on the NextSeq 550 (Illumina).

Sequencing reads were first demultiplexed by index, each library was concatenated and clipped of the molecular modifier added in the second PCR using Je [45]:

java -jar /je_1.2/je_1.2_bundle.jar clip F1 = compiled.gz LEN = 6

Reads were then mapped back to the *Caulobacter crescentus* NA1000 genome (NCBI Reference Sequence: NC_011916.1) using BWA [46] and sorted using Samtools [47]:

bwa mem -t2 clipped.gz | samtools sort -@2 - > sorted.bam

Duplicates were removed using Je [45] and indexed with Samtools [47] using the following command:

java -jar /je_1.2/je_1.2_bundle.jar markdupes I = sorted.bam O = marked.bam M = METRICS.txt MM = 0 REMOVE_DUPLICATES = TRUE

samtools index marked.bam

The 5' insertion site of each transposon were converted into .wig files comprising counts per position and visualized using Integrative Genomics Viewer (IGV) [48,49]. Specific hits for each library were determined with coverage and insertion frequency using a bedfile containing all open reading frames from NC_011916.1 and the outer 20% of each removed to yield a clean and thorough insertion profile. This was determined using BEDTools [50,51] and the following commands:

bedtools genomecov -5 -bg marked.bam > marked.bed

bedtools map -a NA1000.txt -b marked.bed -c 4 > output.txt

Library comparisons were performed using the edgeR package in the Bioconductor suite using a quasi-likelihood F-test (glmQLFit) to determine the false discovery rate adjusted p-values reported here.

Tn-seq data have been deposited in the Sequence Read Archive (SRA) under accession numbers SRX7081604-SRX7081607 and BioProject number PRJNA586876.

## Lipid II extraction and analysis

Precursor extraction was performed as described previously and performed in triplicates [52]. Briefly, 500 mL of WT and $\Delta cdnL$ were grown in PYE to $OD_{600}$ of 0.45. Cells were harvested, resuspended in 5 mL PBS and poured into 50 mL flasks containing 20 mL $CHCl_3$:Methanol (1:2). The mixture was stirred for 1 h at room temperature and centrifuged for 10 min at 4000 x g at 4 °C. The supernatant was transferred to clean 250 mL flasks containing 12 mL $CHCl_3$ and 9 mL PBS, stirred for 1 h at room temperature and centrifuged for 10 min at 4000 x g at 4 °C. The interface fraction (between the top aqueous and bottom organic layers) was collected and vacuum dried. To remove lipid tail, samples were resuspended in 100 μL DMSO and 800 μL $H_2O$, 100 μL ammonium acetate 100 mM pH 4.2 were added. This mixture was boiled for 30 min, dried in vacuum and resuspended in 300 μL $H_2O$. Samples were analyzed by UPLC chromatography coupled to MS/MS analysis, using a Xevo G2- XS QTof system (Waters Corporation, USA). The separation method used is identical to the one used for muropeptide separation explained below. A library of compounds was used to target the identification of peptidoglycan precursors and possible intermediates, although only lipid II (lipid II-penta) was detected in these samples. Lipid II amounts were calculated based on the integration of the peaks (total area), normalized to the culture OD.

## Peptidoglycan (PG) purification and analysis

PG samples were analyzed as described previously [53,54]. Briefly, 50 mL of WT and *ΔcdnL* cells were grown to an $OD_{600}$ of 0.5 in PYE, harvested, and boiled in 5% SDS for 2 h. Sacculi were repeatedly washed with MilliQ water by ultracentrifugation (110,000 rpm, 10 min, 20 ºC) until total removal of the detergent and finally treated with muramidase (100 μg/mL) for 16 hours at 37 ºC. Muramidase digestion was stopped by boiling and coagulated proteins were removed by centrifugation (10 min, 14,000 rpm). For sample reduction, the pH of the supernatants was adjusted to pH 8.5–9.0 with sodium borate buffer and sodium borohydride was added to a final concentration of 10 mg/mL. After incubating for 30 min at room temperature, pH was adjusted to 3.5 with orthophosphoric acid.

UPLC analyses of muropeptides were performed on a Waters UPLC system (Waters Corporation, USA) equipped with an ACQUITY UPLC BEH C18 Column, 130Å, 1.7 μm, 2.1 mm X 150 mm (Waters, USA) and a dual wavelength absorbance detector. Elution of muropeptides was detected at 204 nm. Muropeptides were separated at 45˚C using a linear gradient from buffer A (formic acid 0.1% in water) to buffer B (formic acid 0.1% in acetonitrile) in an 18-minute run, with a 0.25 mL/min flow. Relative total PG amounts were calculated by comparison of the total intensities of the chromatograms (total area) from three biological replicates normalized to the same $OD_{600}$ and extracted with the same volumes. Muropeptide identity was confirmed by MS/MS analysis, using a Xevo G2-XS QTof system (Waters Corporation, USA). Quantification of muropeptides was based on their relative abundances (relative area of the corresponding peak) normalized to their molar ratio.

## Metabolomics sample preparation and analysis

4 mL of each strain was grown to an $OD_{600}$ of 0.3 and filtered through 0.22 μm nylon filters (Millipore GNWP04700). The filter was placed upside down in a 60 mm dish containing 1.2 mL pre-chilled quenching solution (40:40:20 Acetonitrile:methanol:$H_2O$ + 0.5% formic acid) and incubated at -20 ºC for 15 minutes. Cells were washed off the filter by pipetting the quenching solution over the filter, transferred to chilled bead beating tubes containing 50 mg of 0.1 mm glass beads and neutralized with 100 μL 1.9M $NH_4HCO$. Cells were lysed on a bead beater using a Qiagen Tissulyzer at 30Hz for 5 minutes. Samples were spun at 4 ºC for 5 min at max speed and transferred to pre-chilled tubes to remove debris.

LC-MS analysis of cellular extract was conducted on Q Exactive PLUS Hybrid Quadrupole-Orbitrap mass spectrometer (Thermo Fisher Scientific) using hydrophilic interaction chromatography. The Dionex UltiMate 3000 UHPLC system (Thermo Fisher Scientific) with XBridge BEH amide column (Waters, Milford, MA) and XP VanGuard Cartridge (Waters, Milford, MA) were used for LC separation. The LC gradient, comprised of solvent A (95%:5% H2O:acetonitrile with 20 mM ammonium acetate, 20 mM ammonium hydroxide, pH 9.4) and solvent B (20%:80% H2O:acetonitrile with 20 mM ammonium acetate, 20 mM ammonium hydroxide, pH 9.4), corresponded with the following solvent B percentages over time: 0 min, 100%: 3 min, 100%; 3.2 min, 90%; 6.2 min, 90%; 6.5 min, 80%; 10.5 min, 80%; 10.7 min, 70%; 13.5 min, 70%; 13.7 min, 45%; 16 min, 45%; 16.5 min, 100%. Chromatography flow rate was at 300 μL/min and injection volume 5 μL. Column temperature was maintained at 25˚C. MS scans were set to negative ion mode with a resolution of 70,000 at m/z 200, in addition to an automatic gain control target of 3 x 10^6 and scan range of 72 to 1000. Metabolite data was obtained using the MAVEN software package [55].

## Supporting information

**S1 Fig. CdnL levels are unchanged in WT cells grown in PYE, M2G, M2GG and HIGG. A.** Immunoblot against whole cell lysates of the indicated strains (WT (EG865), *ΔcdnL* (EG1447),

*cdnL^{I42N}* (EG1416)) probed with CdnL antisera. **B.** Immunoblot against whole cell lysates of WT cells grown in indicated media. **C.** Quantification of CdnL levels from **B.** using ImageJ. CdnL values are normalized to MreB. Values for each condition are not significantly different from each other using one-way ANOVA with Tukey's multiple comparison test. **D.** Immunoblot against whole cell lysates of Δ*cdnL* P$_{xylX}$-*cdnL* (EG1403) or control Δ*cdnL* P$_{xylX}$-*cfp* (EG3136) strains grown in PYE with xylose (xyl) or glucose (gluc) for the indicated times. CdnL is depleted within 7 hours of growth with glucose in EG1403. MreB was probed as a loading control.
(JPG)

**S2 Fig. Δ*cdnL* and Δ*cdnL*Δ*50kb* have similar phenotypes A.** Growth curves of Δ*cdnL* cells (EG1447) in indicated media. **B.** Phase contrast images of Δ*cdnL*Δ50kb P$_{xylX}$-*cfp* (EG3135) grown in PYE with xylose and imaged at the indicated time points. Time 0 is shown in Fig 2D. **C.** Phase contrast images and **D.** growth curves of WT (EG865), Δ*cdnL* (EG1447) and Δ*cdnL*Δ50kb (EG1415). Bar = 2 µm. **E.** Phase contrast images of Δ*cdnL*Δ50kb complemented with *cdnL* (EG3134) or *cfp* (EG3135) expressed from the P$_{xylX}$ promoter. Cells were grown in PYE xylose, washed, grown in PYE glucose, and imaged at indicated time points. **F.** Growth curve of Δ*cdnL*Δ50kb P$_{xylX}$-*cdnL* and Δ*cdnL*Δ50kb P$_{xylX}$-*cfp* grown in indicated media. Experiments were performed in triplicate and mean is shown.
(JPG)

**S3 Fig. Venn diagram comparing genes at least 2-fold up- or down-regulated in EG1415 versus JC784.**
(JPG)

**S4 Fig. Significantly altered metabolites grown in indicated media from S3 Table. A-D.** WT (EG865) and Δ*cdnL* (EG1447) cells were grown in indicated media until log-phase. **B.** Cells were grown in M2GG, washed and grown in M2G for 12 hours before extracting metabolites. **D.** EG1403 cells were grown without xylose for 8 hours to deplete CdnL before metabolite extraction. P < 0.05.
(JPG)

**S5 Fig. Muropeptide composition is altered in Δ*cdnL* cells. A.** Representative chromatograms of the muramidase-digested sacculi of WT (EG865) and Δ*cdnL* (EG1447) cells. **B.** Relative molar abundance (%) of monomers, dimers, trimers, percentage of crosslinkage (proportion of crosslinked peptide side chains), muropeptides with a residue of (1–6 anhydro) N-acetyl muramic acid (Anhydro), Gly containing muropeptides (Gly muropep.), and pentapeptides. **C.** Muropeptide relative molar abundance (%). GlcNAc: *N*-Acetyl glucosamine. MurNAc: *N*-Acetyl muramic acid. Ala: Alanine. Glu: Glutamic acid. mDAP: meso-diaminopimelic acid. Gly: Glycine. Statistical analysis performed using t-test analysis. * = P < 0.05 and > 10% variation compared to WT.
(JPG)

**S1 Table. RNA-seq and microarray data for EG1415 and JC784, respectively.** Gene (CCNA), annotation, fold change in Δ*cdnL* compared to WT, log$_2$ of fold change, and statistical significance across replicates (1 = significant, 0 = not significant) are presented for each dataset. The 50 kb deletion (*CCNA_02772* to *CCNA_02828*) in EG1415 is highlighted.
(XLSX)

**S2 Table. DAVID [19] functional analysis of the genes at least 2-fold upregulated or 2-fold downregulated in both RNA-seq and microarray datasets from S1 Table.**
(XLSX)

**S3 Table. ChIP-seq data using CdnL or RpoD antibodies.**
(XLSX)

**S4 Table. DAVID [19] functional analysis of the genes identified as putative direct CdnL targets in S3 Table.**
(XLSX)

**S5 Table. Metabolomics data for WT (EG865), Δ*cdnL* (EG1447), Δ*cdnL* P$_{xylX}$::*cdnL* (EG1403) and EG2740 (P$_{xylX}$::*ecfp*) grown in indicated media.**
(XLSX)

**S6 Table. Tn-seq data for Δ*cdnL* (EG1447) and WT (EG865).** The number of unique transposon insertions in each gene in WT compared to Δ*cdnL* was used to determine the log$_2$ fold-change in number of unique transposon insertions (logFC), log counts per million reads (logCPM), P-value, false-discovery rate (FDR), and negative log$_{10}$ of the FDR for each gene in WT versus Δ*cdnL*.
(XLSX)

**S7 Table. Strains and plasmids used in this study and their method of construction.**
(XLSX)

## Acknowledgments

We thank members of the Goley lab for helpful discussions and input. We thank Kousik Sundararajan for identifying the initial mutant of CdnL that suppresses the FtsZΔCTL phenotype. We would like to thank Regis Hallez for providing MreB antisera. We thank the Johns Hopkins Genetic Resources Core Facility for RNA-seq services and analysis, Zach Pincus for guidance in using CellTool and the Metabolomics core at the Rutgers Cancer Institute of New Jersey for metabolomics analysis. We used Biorender.com to create Fig 4B.

## Author Contributions

**Conceptualization:** Selamawit Abi Woldemeskel, Gaël Panis, Diego Gonzalez, Justine Collier, Patrick H. Viollier, Erin D. Goley.

**Data curation:** Selamawit Abi Woldemeskel, Allison K. Daitch, Laura Alvarez, Gaël Panis, Rilee Zeinert, Diego Gonzalez, Erin D. Goley.

**Formal analysis:** Selamawit Abi Woldemeskel, Allison K. Daitch, Laura Alvarez, Gaël Panis, Rilee Zeinert, Diego Gonzalez, Peter Chien, Felipe Cava, Erin D. Goley.

**Funding acquisition:** Justine Collier, Peter Chien, Felipe Cava, Patrick H. Viollier, Erin D. Goley.

**Investigation:** Selamawit Abi Woldemeskel, Allison K. Daitch, Laura Alvarez, Gaël Panis, Rilee Zeinert, Diego Gonzalez, Erika Smith, Erin D. Goley.

**Methodology:** Selamawit Abi Woldemeskel, Laura Alvarez, Gaël Panis, Rilee Zeinert, Diego Gonzalez, Peter Chien, Erin D. Goley.

**Project administration:** Justine Collier, Peter Chien, Felipe Cava, Patrick H. Viollier, Erin D. Goley.

**Resources:** Selamawit Abi Woldemeskel, Rilee Zeinert, Peter Chien, Felipe Cava, Patrick H. Viollier, Erin D. Goley.

**Supervision:** Justine Collier, Peter Chien, Felipe Cava, Patrick H. Viollier, Erin D. Goley.

**Validation:** Selamawit Abi Woldemeskel, Allison K. Daitch, Laura Alvarez, Rilee Zeinert, Diego Gonzalez, Peter Chien, Erin D. Goley.

**Visualization:** Selamawit Abi Woldemeskel, Allison K. Daitch, Laura Alvarez, Gaël Panis, Erika Smith.

**Writing – original draft:** Selamawit Abi Woldemeskel, Allison K. Daitch, Laura Alvarez, Gaël Panis.

**Writing – review & editing:** Selamawit Abi Woldemeskel, Allison K. Daitch, Laura Alvarez, Gaël Panis, Rilee Zeinert, Diego Gonzalez, Erika Smith, Justine Collier, Peter Chien, Felipe Cava, Patrick H. Viollier, Erin D. Goley.

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
