## [Decision Letter · Decision Letter 0]

29 Aug 2019

Dear Dr Goley,

Thank you very much for submitting your Research Article entitled 'The conserved transcriptional regulator CdnL is required for metabolic homeostasis and morphogenesis in Caulobacter' to PLOS Genetics. Your manuscript was fully evaluated at the editorial level and by independent peer reviewers.

The reviewers appreciated the attention to an important problem, but raised some substantial concerns about the current version manuscript.  All three reviewers noted the pleiotropic nature of the cdnL deletion mutant and the need to  perform additional experiment to determine whether particular phenotypes are direct or indirect. For example, reviewer 1 and reviewer 3 requested additional experiments to clarify link between CdnL and FtsZ that was identified in the original suppressor screen. Similarly, reviewer 2 would like more insight into the the molecular basis of the cdnL mutant phenotype and the possible contribution of Lipid II to the same. There were also some technical concerns raised by reviewer 1 about interpretation of the RNAseq and microarray data sets and by reviewer 3 about the need for complementation of the cdnL EG1415 mutant and the large deletion associated with the same.  Based on the reviews, we will not be able to accept this version of the manuscript, but we would be willing to review again a much-revised version. We cannot, of course, promise publication at that time.

We expect that addressing the most significant of the reviewer’s concerns will take a substantial amount of time and understand if you would prefer to resubmit the manuscript to another journal.  Should you decide to revise the manuscript for further consideration here, your revisions should address the specific points made by each reviewer. We will also require a detailed list of your responses to the review comments and a description of the changes you have made in the manuscript.

If you decide to revise the manuscript for further consideration at PLOS Genetics, please aim to resubmit within the next 60 days, unless it will take extra time to address the concerns of the reviewers, in which case we would appreciate an expected resubmission date by email to plosgenetics@plos.org.

[LINK]

We are sorry that we cannot be more positive about your manuscript at this stage. Please do not hesitate to contact us if you have any concerns or questions.

Yours sincerely,

Petra Anne Levin

Reviews Editor

PLOS Genetics

Lotte Søgaard-Andersen

Section Editor: Prokaryotic Genetics

PLOS Genetics

Reviewer's Responses to Questions

**Comments to the Authors:**

Reviewer #1: This is a well-written and clearly presented manuscript describing the effect of deleting cdnL on Caulobacter cell morphology and metabolism. The morphology phenotypes of the cdnL mutants were very thoroughly characterized, and the interpretations tracing the phenotypes back to lipid II and PG limitation makes sense based on the data presented. In general, this manuscript is a thorough and important description of the global implications of loss of this critical transcription factor. There are a few points made by the authors where the data and discussion should be elaborated on to accurately and effectively support their conclusions, as detailed here.

1. The authors state that they became interested in CdnL through a screen for spontaneous suppressors of a dominant lethal mutant in FtsZ that has defects in PG metabolism. This is particularly interesting because of the phenotypes in the �cdnL mutant strain related to PG. However, the authors never tie their findings with the �cdnL mutant strain back to the original suppressor screen. Given what they know now, why do they think the point mutation in cdnL would suppress the FtsZ mutant? This is particularly confounding given that based on the data presented, one would expect that a mutation that destabilizes CdnL should result in defects in PG metabolism and it is not clear how this would suppress the defects in PG metabolism in the FtsZ mutant.

2. Line 206 – When the authors compare their RNA-seq vs. microarray data, they state that there is “significant overlap” between the two datasets. But based on the pie chart in supplementary figure 1, it looks like there are as many genes only differentially expressed in one dataset as there are overlapping. Wouldn’t that suggest that many of the results are specific to one dataset? Does that suggest that the missing 50kbp region has a large effect on the transcriptome? The authors also state they are using a Pearson correlation coefficient for significance, but that’s just indicating that there is a linear relationship. Did the perform a hypergeometric test for over-enrichment between the two datasets?

3. In Figure 1b, it looks like there are more cells in the �cdnL panels undergoing or just finishing division. Is this something that can be quantified? Is there is a delay in the final stages of cell division?

4. It is difficult to interpret the MreB localization data in Figure 2. Does the more mid-cell focused dispersion of MreB along the length of the cell in the CdnL mutants mean that the MreB is mis-localized or does that mean that the cells are simply not elongating due to a lack of PG? Could an alternative explanation of the MreB localization be that more of the CdnL mutant cells are arrested in cell division or have defects in cell cycle progression, leading to persistent MreB localization to mid-cell? Is there a way to differentiate between these possibilities? The authors could analyze FtsZ localization and if the mutant cells are unable to complete fission or are stuck in a particular developmental stage, the proportion of cells with polymerized Z-rings should be different in the mutants. Some discussion of these other interpretations should be included.

The authors could also explain the assumptions that they make when selecting cells for FWHM fair. Can the WT and cdnL cells both be fit to the same Fourier series model? The authors could expand on how they use the demographs to select cells “based on cell length bins”?

5. Related to point 4, it is not clear how CdnL is affecting CtpS abundance or localization following induction of expression from a xylose inducible promoter.

6. Figure 4B could be expanded to be more comprehensive. For example, the authors discuss the glyoxylate bypass in this section of the text, but it is not shown in the figure even though isocitrate lyase and malate synthase are two of the most up-regulated genes.

7. How are the authors normalizing the Lipid II and PG measurements in fig 5? If the measurements are normalized to per cell, could the differences in PG amount simply be due to the smaller size of cdnL cells?

Other comments:

1. The authors should show the data referenced as data not shown in Line 132.

2. The introduction and discussion should be updated to include the recently published expression profiling performed in Mycobacterium tuberculosis strains expressing CarD mutants.

3. Lines 259-260, 264-265 – The authors should map those enzymes onto the figure

4. Figure 5-In fig 5a, the “WT” column seems unnecessary.

5. Fig 5d is a little difficult to read because the black and grey of the WT vs. cdnL points are pretty similar looking.

6. Figure 4- Are the metabolites that are paired above/below in 4A linked in some way? If not, the figure should be presented differently, because it currently the figure implies a relationship between the metabolites.

Reviewer #2: This study by the Goley lab investigates the role of the transcriptional regulator CdnL in metabolism and morphogenesis in Caulobacter. The paper starts with a detailed phenotypic analysis of a cdnL deletion mutant with regard to cell shape, demonstrating that cdnL mutant cells are wider, more curved and have shorter stalks than the wild type. Furthermore, the authors demonstrate that the localization patterns of MreB and CtpS are mildly affected in the cdnL mutant. With the aim to explain the phenotypes of cdnL mutant cells mechanistically, the authors then monitor transcriptome and metabolome changes in the cdnL mutant (in comparison to WT) and search for mutations that are synthetic lethal with the cdnL deletion. These high-throughput analyses indicate that the cdnL mutation has pleiotropic effects and that in particular various metabolic pathways are affected by CdnL loss. Among other effects that are described in the paper, the authors show that cdnL mutant cells have lower levels of Lipid II, which according to the authors could contribute to the observed morphological defects.

Overall, this paper thoroughly describes the phenotypes and cellular consequence of loss of CdnL function. The authors make a number of interesting observations, however, most of these observations remain rather unconnected, leading to only limited mechanistic insight into CdnL functions on morphogenesis. The authors could have performed a number of additional genetic studies to better link their findings. The comments below include suggestions for such experiments as well as a number of other issues that need to be addressed.

Major comments:

1) Introduction: CdnL in Caulobacter has been studied in a previous study by Gallego-Garci et al. The authors should provide more background about the findings that were made in this study. Furthermore, more information about CdnL in other bacteria could be provided in the introduction.

2) The authors compare the cdnL deletion mutant with wild type throughout the study. However, experiments showing that ectopically expressed cdnL can rescue the mutant phenotypes are not included. Given that the cdnL mutant strain E1415 harbors a 50kbp-deletion, such complementation assays are critical to confirm that the observed phenotypes are caused by absence of CdnL.

3) Related to point 2, it is possible that cells lacking CdnL compensate absence of the protein by indirectly up- or downregulating many genes. Indeed, the finding that 30% of the transcriptome is misregulated points to various indirect effects that CdnL loss has on gene expression. To analyse more directly the effects caused by loss of CdnL function the authors could deplete (or replete) CdnL and analyze changes in cell morphology, gene expression and metabolome as a function of time. Furthermore, ChIP-seq experiments would lead to valuable insight into the direct regulon of CdnL, which in turn is expected to lead to a more mechanistic understanding of CdnL function.

4) Fig. 1a: WT Caulobacter cultures seem to grow equally fast in M2G and PYE medium during exponential growth, however, it is well established that the generation time is almost twice as long in M2G compared to PYE medium. The authors should comment on this.

5) Fig. 1a: The growth defects caused by the cdnL deletion are more severe in M2GG and HIGG medium than in PYE. Have the authors compared expression levels of cdnL under the different conditions?

6) Fig. 2: The outcome of the data describing the localization pattern of MreB should be better explained. The authors observe that MreB is hyperfocused in the mutant, but what do these data mean? Is hyperfocused MreB expected to lead to hypercurvature? Futhermore, the data in which the cdnL mutation is combined with different MreB point mutants (Fig. S2) are difficult to understand and the explanations provided by the authors quite vague (l. 166). The authors should try to provide more mechanistic explanations for these observations, or consider to remove these data to avoid confusion.

7) Fig. 3: Similarly to the MreB data, the conclusions regarding CtpS localization data remain unclear. The authors observe that signal intensity of mCherry-CtpS filaments is reduced, but the protein level of CtpS is unchanged (as the authors state in l. 131 based on data not shown). In the discussion, they suggest that CtpS filament formation is impaired in the cdnL mutant as a consequence of lower CTP levels. However, have they ruled out the possibility that mCherry-CtpS levels are lower in cdnL cells compared to WT? Furthermore, it could be tested if modest CtpS overexpression or artificially increased CTP levels can alleviate the hypercurvature phenotype in the cdnL mutant.

8) The authors show that the levels of Lipid II are reduced in the cdnL strain and they suggest that this may contribute to the morphological and cell integrity defects of cdnL mutants. However, experimental data testing the link between decreased lipid II levels and cell morphology are missing. Is it possible to directly assess the contribution of Lipid II misregulation to the morphological phenotypes, for example, by ectopically expressing the genes required for Lipid II synthesis in the cdnL mutant? In an alternative experiment, the authors could test whether artificial downregulation of Lipid II in an otherwise wild type strain phenocopies the morphology of cdnL cells. The authors should also discuss in more detail how reduction in Lipid II may affect the cell wall (l. 292). Is it expected to lead to more or less cross-linking, to longer or shorter polymers, etc? Similarly, the author could elaborate more on possible consequences of changes in the relative amounts of muropeptide species, e.g. Gly-containing muropeptides and pentapeptides (Fig. 5d).

9) The authors explain the observed glutamate auxotrophy of cdnL mutant cells with low amounts of alpha-ketoglutarate and downregulation of gdhA and gltB. However, it remains unclear whether misregulation of glutamate metabolism is linked to the morphological defects or if this is a separate phenotype of cdnL mutants. To address this question it would be interesting to analyze the morphology of cells cultivated in M2G, in which growth is arrested. For example, the authors could shift cells from M2G to M2GG, and then monitor changes in cell morphology over time.

Minor comments:

10) L. 91: “an I42N mutation that disrupted CdnL protein stability”. In Fig. S1a the authors show that CdnL abundance is affected in this mutant, but not necessarily protein stability.

11) L. 123: A citation is missing supporting the statement that stalk elongation in response to phosphate starvation is distinct from developmentally regulated stalk biogenesis.

12) L. 130-131: The experiments probing for changes in protein levels of MreB, CreS, FtsZ and CtpS levels should be shown as they are relevant for the interpretation of the findings shown in Fig. 2&3.

13) L. 178: “and found that cdnL mutant cells form fainter CtpS filaments compared to WT”. Do the authors want to say that the fluorescent signal of localized mCherry-CtpS was fainter than in WT. The wording of this sentence should be modified.

14) L. 180: “mCherry-CtpS forms less robust filaments” What exactly do the authors mean with this statement? Is this really supported by the data?

15) L. 236-237: “The presence of higher uridic acid across all media suggests that d_cdnL cells are preferentially catabolizing amino acids rather than carbohydrates as a carbon source”. The authors should double-check whether the suggested link between uridic acid and amino acid catabolism is correct. To my knowledge, uridic acid is the oxidation product of purine metabolism.

16) L. 274: “that genes required for lipid II biosynthesis” To help the reader, the gene names should be specified.

17) Fig. 1C: Please specify the unit of the y axis in the graphs shown in Fig. 1c.

18) Fig. 4a: Please check that the symbol “alpha” is correctly inserted, e.g. alpha-ketoglutarate, DL-alpha-Glycerophosphate.

Reviewer #3: The PG, shape and division defects have tremendous potential, and a variety of avenues are pursued (that sometimes) reinforce the PG defect model (like the MreB work) and at times takes us on an irrelevant and informative detour (like the CreS/CtpS work), but the most exciting discovery claiming that cdnL-I42N mutation suppresses the growth defect of the ftsZ-CC overexpression mutants is neither explored, nor validated ( I found a cdnL-I42N plasmid in the strain list so I presume it was explored). There is no information about this mutant (how it was obtained), no backcrossing experiments, no description. On the basis of this result the authors embark on a re-evaluation the cdnL deletion phenotype, previously reported to have strong division and shape defects (doi: 10.1038/srep43240).

Importantly, no complementation of the cdnL deletion phenotype is done, and this mutant (EG1415) turns out to have a large deletion elsewhere as was hinted by RNA-seq experiments and then later confirmed. Has any complementation be done to show that phenotypes are caused by the absence of CdnL? All important phenotypes must be tested for their ability to be corrected by providing CdnL in trans in the cdnL deletion mutant background. Were experiments in figure 4 done with EG1415? It should be noted in the figure legends. What about the cell wall analysis in Fig 5 and Fig 6 that were done in EG1415? What are the effects of the 50 kb deletion and has the analysis been done with a complemented mutant? Which mutant was used in Fig S4-S6? The authors have made ΔcdnL, xylX::cdnL (EG1403) strain but there is no mention of it in the text and only used in supplementary Table S3 but the table seems incomplete and the date is not interpreted just dumped as Supplementary table can the authors show a (supplementary) figure illustrating if the complementation actually work and compare it between various cdnL mutant and complemented strains.

The glutamate auxotrophy of the cdnL mutant is novel and exciting, and there seems to be an obvious link with reduced pyruvate levels reported by metabolomics. If so, pyruvate and alpha-ketoglutarate should improve growth of cdnL cells. Is this the case??

Overall, there are some interesting links reported here which I think should be published in PLOS genetics but the major shortcoming is that the work is still very phenomenological and it is not clear how the molecular function of CdnL as general RNAP binding protein can lead to such specific PG and auxotrophic defects. Why are certain genes affected by the absence CdnL? Direct vs indirect effects should be dissected or at least some attempts should be made. For now, this has not been done, and as far as I could tell without complementation analysis described for most phenotypes, it’s not even clear to me what is really caused by inactivation of cdnL versus possible unanticipated (unrelated) genetic defects in the mutant.

The MreB and CtpS work just merely reinforces the notion that something about PG metabolism is going wrong in the cdnL mutant, but why the transcript of PG metabolism is reduced and not MreB and CtpS itself or other genes really remains a mystery. Moreover, the authors don't comment on the FtsZ connection (even in the discussion), despite that there are very obvious models that can be proposed/discussed in the context of their cdnL mutant characterization which would make the story interesting and add a new twist that has not yet been reported. Since the shape and division defects of cdnL were previously already known, this manuscript contributes the metabolism (glutamate auxotrophy) and lipidII defects of the cdnL mutant, if indeed these phenotypes disappear when CdnL returns. This is a solid basis, but the story should be developed.

Additional comments:

The claim that mreB localization is important to cdnL cells has not been shown. The mreB alleles may be partial loss of function (isolated as A22 MPP265 resistant mutants) that also affect localization. There is no causality to be inferred here, localization and activity cannot be separated here. The bottom line is that any stress on the PG pathway will be problematic to cdnL cells, including capsule or other mutations as the authors showed. The MreB and CreS/CtpS parts are not that helpful in understanding the problem with PG precursor biosynthesis.

**Have all data underlying the figures and results presented in the manuscript been provided?**

Reviewer #1: Yes

Reviewer #2: Yes

Reviewer #3: Yes

PLOS authors have the option to publish the peer review history of their article (what does this mean?). If published, this will include your full peer review and any attached files.

Reviewer #1: No

Reviewer #2: No

Reviewer #3: No

---

## [Decision Letter · Decision Letter 1]

1 Jan 2020

Dear Dr Goley,

We are pleased to inform you that your manuscript entitled "The conserved transcriptional regulator CdnL is required for metabolic homeostasis and morphogenesis in Caulobacter" has been editorially accepted for publication in PLOS Genetics. Congratulations and Happy New Year!!

Yours sincerely,

Petra Anne Levin

Reviews Editor

PLOS Genetics

Lotte Søgaard-Andersen

Section Editor: Prokaryotic Genetics

PLOS Genetics

Comments from the reviewers (if applicable):

Reviewer's Responses to Questions

**Comments to the Authors:**

Reviewer #1: The scope and focus of the manuscript has changed a bit when the authors excluded the MreB, CtpS, and CreS data, but overall the authors addressed the reviewers comments adequately, and the result is an important and interesting manuscript.

Reviewer #2: The authors have significantly improved their manuscript based on the reviewers’ suggestions. I have no further comments that need to be addressed.

**Have all data underlying the figures and results presented in the manuscript been provided?**

Reviewer #1: Yes

Reviewer #2: Yes

PLOS authors have the option to publish the peer review history of their article (what does this mean?). If published, this will include your full peer review and any attached files.

Reviewer #1: No

Reviewer #2: No

**Data Deposition**

http://datadryad.org/submit?journalID=pgenetics&manu=PGENETICS-D-19-01152R1

**Press Queries**

---

## [Editor Report · Acceptance letter]

14 Jan 2020

PGENETICS-D-19-01152R1 

The conserved transcriptional regulator CdnL is required for metabolic homeostasis and morphogenesis in *Caulobacter*

Dear Dr Goley, 

We are pleased to inform you that your manuscript entitled "The conserved transcriptional regulator CdnL is required for metabolic homeostasis and morphogenesis in *Caulobacter*" has been formally accepted for publication in PLOS Genetics! Your manuscript is now with our production department and you will be notified of the publication date in due course.

With kind regards,

Matt Lyles

PLOS Genetics

On behalf of:
